# Wavy Transformer

**Satoshi Noguchi**
Research Institute for Value-Added Information Generation, JAMSTEC
Center for Advanced Intelligence Project, RIKEN
satoshin@jamstec.go.jp

**Yoshinobu Kawahara**
Graduate School of Information Science and Technology, The University of Osaka
Center for Advanced Intelligence Project, RIKEN
kawahara@ist.osaka-u.ac.jp

## Abstract

Transformers have achieved remarkable success across natural language processing (NLP) and computer vision (CV). However, deep transformer models often suffer from an over-smoothing issue, in which token representations converge to similar values as they pass through successive transformer blocks. In this paper, we establish an equivalence between the hidden-state dynamics induced by stacked attention layers and graph neural diffusion on a complete graph. From this perspective, over-smoothing can be interpreted as a consequence of the dissipative nature of the underlying diffusion dynamics. Motivated by this physical interpretation, we propose Wavy Transformer, which consists of a novel attention layer based on second-order wavy dynamics. We also introduce a feed-forward network and a normalization layer designed to preserve the physical state-velocity relationship under the chain rule, thereby extending the transformer architecture. We further validate our proposed techniques on various transformer models for NLP, CV, and sparse-graph tasks. The results consistently demonstrate that Wavy Transformer improves performance with minimal additional parameters and no extra hyperparameter tuning. Source code and models are available at https://github.com/noguchisatoshi/Wavy-Transformer.

## 1 Introduction

Transformers [41] have achieved outstanding success in a wide range of machine learning fields such as natural language processing (NLP) [41, 11, 7, 3] and computer vision (CV) [39, 13, 26, 1, 4, 24]. These successes clearly demonstrate the effectiveness and generality of the transformer. Despite this remarkable progress, deep transformer models often suffer from an over-smoothing issue, in which all token representations become identical as more layers are added [12, 29, 43, 37]. This phenomenon is also known as the token uniformity problem [12] and is recognized as a crucial obstacle preventing transformers from going deeper. Accordingly, several techniques to mitigate the over-smoothing behavior of transformers have been proposed [37, 43, 29]. However, compared to graph neural networks (GNNs), where over-smoothing was first identified and widely studied [34, 31, 28, 14, 35, 38, 20, 30, 2, 8, 17], over-smoothing in transformers has not been adequately discussed.

Based on the above discussion, in this paper, we examine over-smoothing in transformers from the perspective of physical dynamical systems. In addition, we propose a new type of attention layer inspired by physical dynamical systems. Our approach originates from interpreting the dynamics of hidden states induced by stacked attention layers as graph neural diffusion [5] on a complete

39th Conference on Neural Information Processing Systems (NeurIPS 2025).

graph. From this viewpoint, over-smoothing can be seen as a consequence of the dissipative nature of the underlying diffusive dynamics. Motivated by this physical interpretation, we introduce Wavy Transformer, which consists of a novel attention layer based on second-order wavy dynamics. The energy-preserving property and the oscillatory behavior of the wave equation are expected to help mitigate the over-smoothing problems in transformers. We also introduce a physically inspired feed-forward network and a normalization layer designed to preserve the state-velocity relationship under the chain rule. By combining these components, we build Wavy Transformer to extend the conventional transformer architecture. Wavy Transformer block is easy to use and can be integrated into various transformer-like architecture. Furthermore, to demonstrate the effectiveness and generality of Wavy Transformer, we conduct extensive experiments on NLP, CV, and sparse-graph tasks.

The remainder of this paper is organized as follows. In Section 2, we provide background on this work, including an introduction to the attention module and its over-smoothing issue, as well as a brief review of the fundamental properties of dynamics on smooth manifolds. In Section 3, we present Wavy Transformer, after discussing a rigorous physical interpretation of attention layers as graph neural diffusion on a complete graph. In Section 4, we review related works. In Section 5, we report experimental results demonstrating the capabilities of Wavy Transformer. Finally, we conclude with our main contributions and some closing remarks in Section 6.

## 2 Background

### 2.1 Attention and Over-smoothing

The key functional component of transformer architecture is the attention module [41, 43] which aggregates information from other token representations with respect to the computed attention values. Let $\mathbf{X} \in \mathbb{R}^{n \times d}$ be the input states to an attention layer, where $n$ is the number of input tokens and $d$ is the embedding dimension. The attention calculation is formulated as follows:

$$\text{Attn}(\mathbf{X}) = \text{softmax}\left( \frac{\mathbf{X}\mathbf{W}_Q(\mathbf{X}\mathbf{W}_K)^\top}{\sqrt{d}} \right) \mathbf{X}\mathbf{W}_V = \mathbf{A}\mathbf{X}\mathbf{W}_V, \qquad (1)$$

where $\mathbf{W}_K \in \mathbb{R}^{d \times d_k}$, $\mathbf{W}_Q \in \mathbb{R}^{d \times d_k}$ and $\mathbf{W}_V \in \mathbb{R}^{d \times d}$ are the key, query and value weight matrices, respectively. Also, $\sqrt{d}$ denotes a scaling factor, and $\text{softmax}(\cdot)$ operation on $\mathbf{X}$ row-wisely. Here, $\mathbf{A} := \text{softmax}\left( \frac{\mathbf{X}\mathbf{W}_Q(\mathbf{X}\mathbf{W}_K)^\top}{\sqrt{d}} \right) \in \mathbb{R}^{n \times n}$. In the case of Multi-Head Attention (MHA) [41], multiple single-head attention modules operate in parallel, and their outputs are concatenated and linearly projected: $\text{MHA}(\mathbf{X}) = [\text{Attn}_1(\mathbf{X}), \cdots, \text{Attn}_H(\mathbf{X})]\mathbf{W}_O$, where the subscripts denote the head indices, $H$ is the total number of heads, and $\mathbf{W}_O \in \mathbb{R}^{Hd \times d}$ projects the concatenated outputs back to the hidden dimension. Since MHA is a parallelized form of single-head attention and is mathematically equivalent, we do not distinguish between them in this paper. In addition to the attention module, each transformer block is equipped with residual connections as:

$$\mathbf{X}^{l+1} = \mathbf{A}\mathbf{X}^l\mathbf{W}_V + \mathbf{X}^l, \qquad (2)$$

where $\mathbf{X}^l$ denotes the $l$-th layer hidden states. The attention matrix can be regarded as learning pair-wise self-interactions among all components of $\mathbf{X}$. Thus, this attention computation encourages transformers to capture globally direct interactions among all tokens, unlike convolutional operations, which expand their receptive field hierarchically from local to global. As a consequence, properly trained transformers can effectively model global context and achieve outstanding performance.

Despite their remarkable performance across a wide range of NLP and CV applications, deep transformer models often suffer from an over-smoothing problem: as the network deepens, token representations converge and become indistinguishable from one another [37, 43, 29]. As a result, deeper transformers do not always outperform their shallower counterparts [49]. From a theoretical viewpoint, the attention layer can be understood as a low-pass filter [43], continuously erasing high-frequency information and thus reducing feature expressiveness in deeper layers. Consequently, this tendency toward over-smoothing of stacked the attention layers in deep transformer presents a critical challenge that motivates our study.

## 2.2 Fundamental Dynamics on Manifolds

Here, we briefly review classical dynamics on a smooth manifold $M$, described by partial differential equations (PDEs), to motivate alternative dynamics for the transformer block [15]. For simplicity, we omit any specification of initial or boundary conditions whenever it is not necessary. Let $x : M \times [0, T] \to \mathbb{R}$ denote a scalar field on $M$, where $T > 0$ is the terminal time. Its prototypical example is the diffusion equation:

$$\frac{\partial x(s, t)}{\partial t} = \mathrm{div}\big(D \,\nabla x(s, t)\big) = D \,\Delta x(s, t), \tag{3}$$

where $s \in M$ denotes a point (spatial coordinate) on the manifold, $t \in [0, T]$ denotes time, $D \geq 0$ is a diffusivity coefficient, $\nabla$ and $\mathrm{div}$ are respectively the gradient and divergence operators induced by the Riemannian metric on $M$, and $\Delta = \mathrm{div} \circ \nabla$ is the Laplace–Beltrami operator. Here, $D$ is assumed to be constant for simplicity. In physics, Eq.(3) models processes such as heat conduction or particle diffusion. In contrast, the hyperbolic (wave) equation on $M$ takes the form

$$\frac{\partial^2 x(s, t)}{\partial t^2} = \mathrm{div}\big(v^2 \,\nabla x(s, t)\big) = v^2 \,\Delta x(s, t), \tag{4}$$

where $v \geq 0$ is a constant wave-propagation speed. Equation (4) governs wave-like phenomena in continuous media. We highlight the key differences between the dynamics of these PDEs.

### 2.2.1 Kernel Perspective of Diffusion and Wave Equations

Fundamental solutions, which are often called "kernels", represent the response to an initial impulse (e.g., a Dirac delta function). These kernels underscore some of the most striking differences between diffusion and wave phenomena. Thus, we review the kernels of both the diffusion and wave equations to clarify how they differ. For simplicity, we focus on the one-dimensional case in the reminder of this section. We also provide fundamental theorems from an energetic perspective that characterize each dynamic in Appendix C.

**Diffusion Equation** We recall the fundamental solution (Green's function) of the one-dimensional diffusion equation [15] with the initial condition $x(s, 0) = x_0(s)$. Its kernel (Green's function) on $\mathbb{R}$ is given by

$$G(s, t) = \frac{1}{\sqrt{4\pi t}} \exp\!\left(-\frac{s^2}{4t}\right), \quad t > 0, \tag{5}$$

where we ignore the diffusive coefficient $D$. Accordingly, the solution with initial data $x_0$ can be written in convolution form: $x(s, t) = \int_{-\infty}^{\infty} G(s - v, t)\, x_0(v)\, \mathrm{d}v$. Key properties of $G$ in one dimension are as follows: Firstly, $G(s, t) > 0$ for all $s \in \mathbb{R}$ and decays exponentially as $|s| \to \infty$. And secondly, the heat kernel smooths out irregularities in the initial data, which is referred to as the *smoothing effect*. The first property shows that, while the diffusive influence extends over all space, it becomes exponentially small far from the origin. The second property implies that diffusion rapidly weakens high-frequency components, inherently suppressing local structural details.

**Wave Equation** Next, we consider the one-dimensional wave equation with initial conditions $x(s, 0) = f(s)$ and $\frac{\partial x}{\partial t}(s, 0) = g(s)$. By d'Alembert's formula [32], the general solution is

$$x(s, t) = \frac{1}{2}\big[f(s - v\,t) + f(s + v\,t)\big] + \frac{1}{2v} \int_{s - v\,t}^{s + v\,t} g(r)\, \mathrm{d}r, \tag{6}$$

which represents left- and right-traveling waves emanating from the initial data. Key properties of this solution are as follows: Firstly, under wave propagation, strong singularities in the initial data can persist. Thus, the signal profile propagates at finite speed, rather than being smoothed out everywhere. And secondly, the general solution reflects conservation of energy: waves transport energy through the domain without diffusive spreading. Hence, in contrast to the diffusion equation, the wave equation exhibits a sharp, finite wavefront traveling at speed $v$. This difference in how features propagate is closely related to the fact that the wave equation conserves energy, whereas the diffusion equation dissipates it. The energy behaviors of each dynamics are shown in Appendix C.

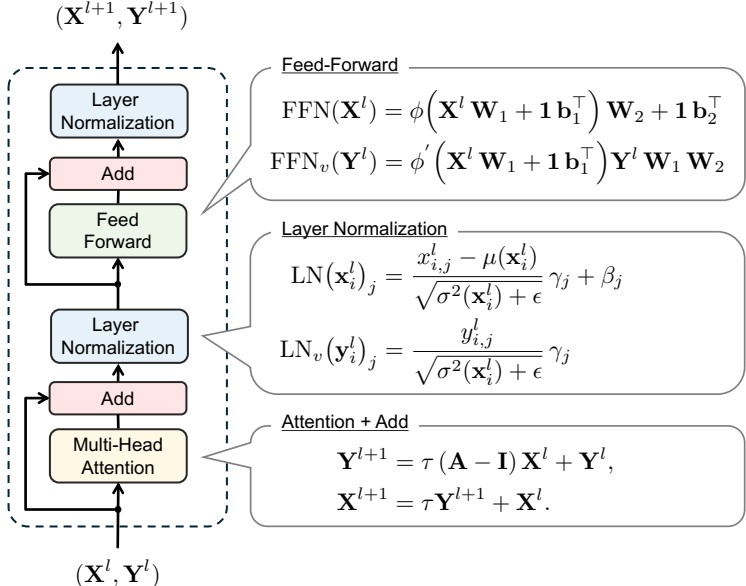

Figure 1: Schematic of Wavy Transformer block, combining wavy attention layers and velocity-specific layer-normalization and feed-forward layers. Each layer is designed to preserve state-velocity relationship. Post-LN is assumed here; the Pre-LN case is discussed in Appendix D.2 [47].

## 3 Wavy Transformer

The above discussion and the fundamental theorems 1 and 2 in Appendix C indicate that the dynamics governed by the diffusion equation (3) converge to a constant uniform state, whereas those governed by the wave equation (4) oscillates while conserving its energy without net dissipation. However, conventional transformers implicitly rely on the dissipative diffusion dynamics, causing their hidden states to converge toward a uniformly constant state. Therefore, before presenting the architecture of Wavy Transformer, we first show that the implicit dynamics of attention layers can be interpreted as graph neural diffusion [5] on a complete graph.

### 3.1 Attention as Graph Neural Diffusion

We consider the graph neural diffusion [5] on a complete graph whose each node feature is token representations:

$$\frac{\partial \mathbf{X}}{\partial t} = (\mathbf{A} - \mathbf{I})\mathbf{X}, \tag{7}$$

where $\mathbf{A}$ is the attention matrix and $\mathbf{I}$ is the identity matrix. For simplicity, we temporarily ignore the feature transformation by $\mathbf{W}_V$. By definition, $\mathbf{A}$ is a right-stochastic matrix, i.e. $\sum_j \mathbf{A}_{ij} = 1$. Hence, $\mathbf{I} - \mathbf{A}$ can be regarded as the normalized graph Laplacian of the complete graph in which the edge connecting nodes $i$ and $j$ is weighted by the node-pair value $\mathbf{A}_{ij}$. Note that $\mathbf{I} - \mathbf{A}$, which is often called the random-walk graph Laplacian [21], is the negative of the discrete Laplacian used here. We can also discretize this graph diffusion equation in time domain as

$$\mathbf{X}^{l+1} = \tau(\mathbf{A} - \mathbf{I})\mathbf{X}^l + \mathbf{X}^l = \tau\mathbf{A}\mathbf{X}^l + (1 - \tau)\mathbf{X}^l, \tag{8}$$

for $l = 1, \cdots, N$, where $N$ is the number of layers and $\tau > 0$ is a fixed time interval parameter. Also, $\mathbf{X}^l$ denotes the hidden states at time $l\tau$. Interestingly, if we consider $\tau = \frac{1}{2}$, we can get $\mathbf{X}^{l+1} = \frac{1}{2}(\mathbf{A}\mathbf{X}^l + \mathbf{X}^l)$. Usually, since each attention layer is followed by a layer normalization (LN), the scale transformation can be ignored. Therefore, this discrete diffusion equation is essentially identical to the attention layer update $\mathbf{X}^{l+1} = \mathbf{A}\mathbf{X}^l + \mathbf{X}^l$. Although we here assume Post-LN [47], in the same way, the attention with Pre-LN can be understood as diffusion–reaction equation, which is explained in Appendix D. In this sense, the dynamics of attention layers can be understood as a graph diffusion on a complete graph. Furthermore, because diffusion decreases the system's energy (as shown in Appendix C), this interpretation aligns with the discussion in [44, 29].

As a result, the conventional transformer architecture can be viewed as executing a diffusion process on a complete graph in which each node represents a token embedding. Moreover, the attention matrix $\mathbf{A}$ defines anisotropic diffusivity interactions among tokens. This perspective suggests that the over-smoothing observed in transformers results from the dissipative smoothing effect of a diffusion equation, analogous to heat flow. An intuitive explanation of this phenomenon is provided in Appendix E. By viewing diffusion as a process that smooths small-scale fluctuation erasing high-frequency components, it becomes clear why the attention layer acts as a low-pass filter [43]. It also agrees with the interpretation of GNNs as performing low-pass filtering, as noted in [28].

In the field of GNNs, there are some attempts to define networks based on the wave equation for mitigation of their over-smoothing issues, focusing on its property of energy conservation [14, 35]. Considering the finding that transformer architecture can be regarded as the graph neural diffusion on a complete graph, introducing a wavy dynamics into the transformer is also expected to enhance its performance. Therefore, this paper introduces a novel type of transformer block based on the wavy dynamics and validates its effectiveness and generality several experiments.

### 3.2 Wavy Dynamics Based Attention

We introduce a novel transformer block based on wavy rather than diffusive dynamics, motivated by the implicit reliance of conventional transformers on diffusive dynamics and some discussion for GNNs [14, 35]. Figure 1 shows a schematic of Wavy Transformer block, which includes a wavy attention layer, velocity-specific layer normalization, and a velocity-oriented feed-forward network. We begin by considering the wave equation on a complete graph, analogous to the graph neural diffusion [5]:

$$\frac{\partial^2 \mathbf{X}}{\partial t^2} = (\mathbf{A} - \mathbf{I})\,\mathbf{X}. \tag{9}$$

By introducing the artificial *velocity* variable $\mathbf{Y} = \frac{\partial \mathbf{X}}{\partial t}$, we can rewrite the second-order wave equation as a first-order systems:

$$\frac{\partial \mathbf{Y}}{\partial t} = (\mathbf{A} - \mathbf{I})\,\mathbf{X}, \quad \frac{\partial \mathbf{X}}{\partial t} = \mathbf{Y}. \tag{10}$$

Here, $\mathbf{Y}$ can be considered as a difference of hidden states across adjacent layers. The following update rule can be obtained by time-discretization of the system Eq.(10):

$$\mathbf{Y}^{l+1} = \tau\,(\mathbf{A} - \mathbf{I})\,\mathbf{X}^l + \mathbf{Y}^l, \quad \mathbf{X}^{l+1} = \tau\mathbf{Y}^{l+1} + \mathbf{X}^l. \tag{11}$$

for $l = 1, \cdots, N$, where $N$ is the number of layers and $\tau > 0$ is a fixed time interval parameter. Also, $\mathbf{X}^l$ and $\mathbf{Y}^l$ denote the hidden states at time $l\tau$. Also, importantly this discrete dynamical system is symplectic, meaning it conserves its system energy [48]. Since this property might contribute to avoid the over-smoothing behavior of the hidden states $\mathbf{X}$, we consider the wave transformer block based on Eq.(10) instead of Eq.(9)

To highlight the difference from the basic diffusive attention layer, if we directly discretize Eq.,(9) in time, rather than bypassing it through Eq.,(10), we obtain $\mathbf{X}^{l+1} = \tau^2\mathbf{A}\mathbf{X}^l + (1 - \tau^2)\mathbf{X}^l + (\mathbf{X}^l - \mathbf{X}^{l-1})$. Ignoring the difference in the constant coefficient ($\tau^2$), compared to the purely diffusive update rule, this wavy update rule includes an additional term, $(\mathbf{X}^l - \mathbf{X}^{l-1})$ which represents a change in the hidden states similar to velocity. In other words, this can be understood as incorporating momentum into the update process. By preserving past state changes, this mechanism prevents excessive smoothing of node features across layers, allowing richer information to propagate and mitigating over-smoothing.

Also, combinations of first-order diffusion and second-order wave dynamics can be employed. One option is to blend the outputs of the conventional (diffusive) attention layer and the proposed wavy attention layer: $\mathbf{X}^{l+1} = \boldsymbol{\lambda}\,\mathbf{X}^{l+1}_{\text{wave}} + (1 - \boldsymbol{\lambda})\,\mathbf{X}^{l+1}_{\text{diffuse}}$, where $\mathbf{X}^{l+1}_{\text{wave}}$ and $\mathbf{X}^{l+1}_{\text{diffuse}}$ are given by Eqs. (11) and (8), respectively. The mixing vector $\boldsymbol{\lambda} \in [0,1]^d$ is defined as $\boldsymbol{\lambda} = \text{sigmoid}(\boldsymbol{\theta})$, with $\boldsymbol{\theta} \in \mathbb{R}^d$ a trainable parameter vector. Alternatively, the velocity update in Eq. (11) can be replaced by $\mathbf{Y}^{l+1} = \boldsymbol{\lambda}\big[\tau(\mathbf{A} - \mathbf{I})\mathbf{X}^l + \mathbf{Y}^l\big] + (1 - \boldsymbol{\lambda})(\mathbf{A} - \mathbf{I})\mathbf{X}^l$. Although $\mathbf{Y}$ is somewhat physically unnatural due to mixing dimension-inconsistent two terms, we can regard it pragmatically as the layer-to-layer state change by combining diffusive and wavy components controlled by $\boldsymbol{\lambda}$.

## 3.3 Physically Consistent Layer Normalization and Feed-forward Network

Beside attention module, each transformer block is equipped with a layer normalization and a feed-forward network. We consider extension of each layer for preserving the physical state-velocity relationship $\mathbf{Y} = \frac{\partial \mathbf{X}}{\partial t}$.

### 3.3.1 Layer Normalization for Velocity

Layer normalization is formally defined for each $d$-dimensional feature vectors $\mathbf{x}_i^l = \left[ x_{i,1}^l,\ x_{i,2}^l,\ \ldots,\ x_{i,d}^l \right]$ included in $l$-th layer's hidden state $\mathbf{X}^l$ as

$$\mathrm{LN}\bigl(\mathbf{x}_i^l\bigr)_j = \frac{x_{i,j}^l - \mu(\mathbf{x}_i^l)}{\sqrt{\sigma^2(\mathbf{x}_i^l) + \epsilon}}\, \gamma_j + \beta_j \quad (i = 1, 2, \ldots, n,\ j = 1, 2, \ldots, d), \tag{12}$$

where $\mu(\mathbf{x}_i^l) = \frac{1}{d} \sum_{j=1}^d x_{i,j}^l$, $\sigma^2(\mathbf{x}_i^l) = \frac{1}{d} \sum_{j=1}^d \left( x_{i,j}^l - \mu(\mathbf{x}_i^l) \right)^2$. And $\gamma_j$ and $\beta_j$ are learnable parameters, and $\epsilon$ is a small constant for numerical stability. Considering the state-velocity relationship $\mathbf{Y} = \frac{\partial \mathbf{X}}{\partial t}$, we can define the corresponding layer romanization for $d$-dimensional velocity vector $\mathbf{y}_i^l = \left[ y_{i,1}^l,\ y_{i,2}^l,\ \ldots,\ y_{i,d}^l \right]$ included in $\mathbf{Y}^l$ as follows:

$$\mathrm{LN}_v\bigl(\mathbf{y}_i^l\bigr)_j = \frac{y_{i,j}^l}{\sqrt{\sigma^2(\mathbf{x}_i^l) + \epsilon}}\, \gamma_j \quad (i = 1, 2, \ldots, n,\ j = 1, 2, \ldots, d). \tag{13}$$

Based on the relationship, only the scaling parameters (e.g., $\sigma^2(\mathbf{x}_i^l)$ and $\gamma_j$) are applied, while the shit parameters (e.g., $\mu(\mathbf{x}_i^l)$ and $\beta_j$) are ignored. In addition, velocity vectors are normalized using mean and variance of states $\mathbf{X}^l$ as well.

### 3.3.2 Feed-forward Network for Velocity

A typical feed-forward network used in transformer architectures is a two-layer fully connected network. Basically, we can define the FFN output as follows:

$$\mathrm{FFN}(\mathbf{X}^l) = \phi\Bigl(\mathbf{X}^l\,\mathbf{W}_1 + \mathbf{1}\,\mathbf{b}_1^\top\Bigr)\mathbf{W}_2 + \mathbf{1}\,\mathbf{b}_2^\top, \tag{14}$$

where $W_1 \in \mathbb{R}^{d \times d}$ and $W_2 \in \mathbb{R}^{d \times d}$ are the weight matrices, $\mathbf{b}_1 \in \mathbb{R}^d$ and $\mathbf{b}_2 \in \mathbb{R}^d$ are the bias vectors, and $\mathbf{1} \in \mathbb{R}^n$ is a vector of ones (used here to broadcast the biases across all rows). Also, $\phi(\cdot)$ is a nonlinear activation function (often ReLU or GELU). Applying this operation transforms each row of $\mathbf{X}^l$ (the hidden state of each token) through a two-layer MLP with a nonlinear activation, resulting in an output matrix of the same size ($n \times d$).

In the same way as the layer normalization, we can straightforwardly define a feed-forward network for velocity according to the chain rule as follows:

$$\mathrm{FFN}_v(\mathbf{Y}^l) = \phi^{'}\Bigl(\mathbf{X}^l\,\mathbf{W}_1 + \mathbf{1}\,\mathbf{b}_1^\top\Bigr)\mathbf{Y}^l\,\mathbf{W}_1\,\mathbf{W}_2, \tag{15}$$

where $\phi^{'}$ represents the derivative function of the activation function $\phi$. Importantly, the velocity is scaled not only by the weight parameters (e.g., $\mathbf{W}_1$ and $\mathbf{W}_2$) but also by the derivative function $\phi^{'}$.

# 4 Related Works

## 4.1 Mitigating Over-Smoothing: Existing Approaches and This Study

To alleviate the over-smoothing problem, some techniques have been introduced in the fields of GNNs [35, 14, 45, 34] and transformer-based models [43, 29, 28]. From the viewpoint of dynamical systems, these approaches can be summarized as some attempts to inject high-frequency disturbance into hidden states dynamics to prevent over-smoothing. They can be broadly classified into two main categories: (i) direct injection of high-frequency signals as an external disturbance to prevent convergence toward uniform hidden states (e.g. [43, 29]), and (ii) introduction of intrinsic modifications of the governing dynamical system that propagate high-frequency modes within the hidden states (e.g. [35, 14]). Almost all existing attempts to alleviate the over-smoothing problem in transformer-based models belong to category (i). Therefore, this paper offers the first comprehensive discussion of an intrinsic mechanism to prevent over-smoothing in transformer-based networks.

### 4.2 Physics-Inspired GNN and Transformers

Our study is complementary to prior physics-inspired GNNs and transformers. Graph-CON [35] and PDE-GCN [14] introduce oscillatory or PDE-motivated updates on sparse graphs; in contrast, we target complete-graph attention in transformers. Relatedly, Gravina et al.[18] analyze over-squashing in sparse-graph GNNs, whereas we focus on over-smoothing in all-to-all attention. Deng et al.[9] introduce Hamiltonian structure into transformers through a loss, while we directly replace the residual dynamics with a second-order wave update without auxiliary losses (see also HNNs [19]).

### 4.3 Baselines for Comparative Study

In our comparative study, we include BERT, DeiT, and DIFFormer as representative baselines for NLP, CV, and sparse-graph applications, respectively.

**BERT: Bidirectional Encoder Representations from Transformers** BERT, introduced by Devlin et al. [11], is a deep bidirectional transformer encoder trained with masked language modeling and next sentence prediction objectives. By conditioning on both left and right context in every layer, BERT learns rich contextual embeddings that serve as a standard benchmark for a wide range of NLP tasks. Thus, we chose BERT as the canonical NLP model for comparison.

**DeiT: Data-efficient Image Transformers** DeiT, proposed by Touvron et al. [39], adapts the transformer architecture to vision tasks without massive pre-training. DeiT exemplifies a state-of-the-art CV transformer, which we use as our baseline model for CV task. Also, we review an existing over-smoothing mitigation technique [43]: FeatScale, which re-weights features to boost high-frequency signals. We then show Wavy Transformer can seamlessly integrate with it to enhance their performance.

**DIFFormer: Diffusion-based Graph Transformers** As a representative graph transformer, we adopt DIFFormer [44] as our diffusion baseline on sparse-graph benchmarks.

## 5 Experiments

In this section, we present a comprehensive experimental evaluation of Wavy Transformer block for NLP tasks under a BERT-like pretraining framework [11], measuring (i) pretraining perplexity (PPL) and MLM accuracy, (ii) fine-tuning performance on GLUE tasks [42], and (iii) over-smoothing behavior. Moreover, to demonstrate the generality and effectiveness of our approach with other types of transformer-based backbones DeiT [39], we validate Wavy Transformer on representative computer vision tasks, specifically ImageNet classification [10]. Also, we evaluated the performance of Wavy Transformer on sparse-graph benchmark based on DIFFormer [44]. Experiments used servers with eight V100 GPUs, four L40 GPUs, or four H100 GPUs, selected according to task size.

**Model Variants** We evaluate two variants of the Wavy Transformer. Both share the same attention backbone; they differ only in how the velocity residual is realized. **Full Wave.** A second-order residual realized with an explicit velocity branch that includes a FFN and a LN on the velocity path as shown in Figures 1, yielding stronger physical constraints at additional computational costs. **Light Wave.** A momentum-only realization that keeps the second-order effect while removing FFN/LN for velocity. The update is $\mathbf{X}^{l+1} = \tau \mathbf{A} \mathbf{X}^l + (1 - \tau)\mathbf{X}^l + \boldsymbol{\lambda}(\mathbf{X}^l - \mathbf{X}^{l-1})$, where $\boldsymbol{\lambda} \in [0, 1]^d$ controls the mix between the diffusion part and the wave (momentum) part. Ignoring the coefficient mismatch between the diffusion term ($\tau$) and the momentum term ($\tau^2$), setting $\boldsymbol{\lambda} = \mathbf{0}$ recovers diffusion-only, whereas $\boldsymbol{\lambda} = \mathbf{1}$ corresponds to the wave-only residual. Formal derivations appear in Appendix. B.

### 5.1 Experiments: NLP Tasks

Here, we adopt a smaller-scale pretraining setup than standard BERT configurations to facilitate faster experimentation. The implementation details are given in Appendix A.1.1

#### 5.1.1 PPL and MLM Accuracy

First, we present the pretraining results to illustrate the fundamental capacity of each model with its respective residual connection as a language model. Table 1 summarizes the validation PPL and

Table 1: Validation PPL and MLM accuracy (MLM Acc) after 10k steps. Arrow symbols denote that lower PPL and higher MLM Acc are better.

| Residual | PPL ($\downarrow$) | MLM Acc ($\uparrow$) |
|---|---|---|
| Diffusion | 31.76 | 44.39 % |
| Full Wave | 31.99 | 44.52 % |
| Mix (+Full) | **29.00** | **45.56 %** |
| Mix (+Light) | 32.29 | 44.53 % |

Table 2: GLUE results for various models. The highest score for each task is highlighted in bold.

| Residual | CoLA | SST-2 | MRPC | QQP | MNLI-m/-mm | QNLI | RTE | WNLI | STS-B | Avg. |
|---|---|---|---|---|---|---|---|---|---|---|
| Diffusion | 10.67 | 83.64 | **76.36** | 83.96 | 73.51/**74.03** | 81.63 | 52.95 | 51.11 | 52.11 | 64.13 |
| Full Wave | 10.64 | 83.10 | 76.11 | **84.17** | 72.10/72.94 | 80.74 | 53.67 | **56.34** | 32.91 | 62.27 |
| Mix (+Full) | 12.07 | **84.25** | 76.32 | 83.81 | 73.04/73.53 | 81.19 | 55.35 | 55.40 | 29.40 | 62.44 |
| Mix (+Light) | **12.77** | 82.76 | 76.18 | 83.89 | **73.71**/73.77 | **81.98** | **55.96** | 55.40 | **64.76** | **66.12** |

MLM accuracy after 10k steps of batch size 64 for each model. Interestingly, while the wavy residual connection alone just achieves nearly the same-level performance as the diffusive residual connection, the mixed residual connections outperform both. This suggests that the wavy residual connection capture fundamentally different or complementary dynamics compared to the usual diffusive residual connection, and that gating both types of updates together with a trainable $\lambda$ enables the model to leverage a broader range of interactions than a purely diffusive attention update.

### 5.1.2 GLUE Fine-tuning Performance

The GLUE [42] benchmark provides a comprehensive suite of natural language understanding tasks that serve as critical tests for evaluating the performance of pretrained language models. In our experiment, we fine-tuned each pretrained BERT-like model on each GLUE task dataset for demonstrating advantages of Wavy Transformer over baseline methods. As summarized in Table 2, the mixed residual with the momentum-only variant (Mix with Light Wave) attains the best macro average over the 10 metrics (+1.99 over Diffusion). Gains are concentrated on CoLA, QNLI, RTE, WNLI, and especially STS-B (+12.65 vs. Diffusion), while remaining competitive on MNLI-m; minor regressions appear on SST-2, MRPC, QQP, and MNLI-mm. The Mix with Full Wave variant reaches the top SST-2 but underperforms overall due to a large STS-B drop with the corrected evaluator. Together with Table 1, these results strongly suggest (i) that Wavy Transformer can capture different features unattainable by conventional diffusive connections, and (ii) that combining these different types of features can enhance BERT's performance on NLP tasks.

### 5.1.3 Analysis of Over-smoothing

We further conduct the analysis of the over-smoothing behavior for each residual connections. Especially, we computed the cosine similarity as a measure to investigate the over-smoothing behavior following [37]: $\text{CosSim} = \frac{1}{n(n-1)} \sum_{i \neq j} \frac{\mathbf{x}_i^\top \mathbf{x}_j}{\|\mathbf{x}_i\| \|\mathbf{x}_j\|}$. For computation of the cosine similarity, we use data randomly sampled from WIkipedia and BookCorpus as input to the BERT-based pretraining models fine-tuned on the SQuAD dataset [33]. Figures 2–3 compare layerwise cosine similarity: Figure 2 contrasts the difference between diffusive and wavy, and Figure 3 shows the mixed residual with Full Wave (with $1\sigma$ shading). As seen in Figure 2, diffusive and wavy residuals exhibit similar over-smoothing at the 24th layer (means and variances are comparable), but their trajectories markedly differ: diffusion increases monotonically with depth, whereas wave oscillates and drops near the final layer, consistent with wave-equation updates. This suggests that diffusive and wavy residual connections extract fundamentally different types of features from the data, which may be related to the frequency of hidden states across layers. Figure 3 shows the mixed residual; together with Tables 1 and 2, this supports that wavy features complement diffusive ones, and that gating them broadens the feature range and improves performance.

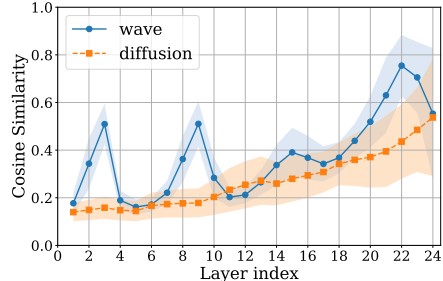 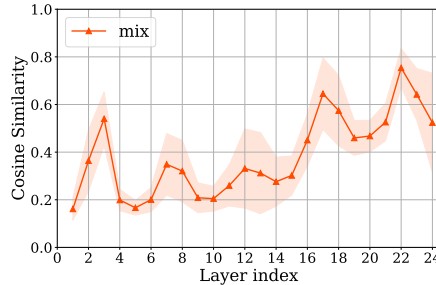

Figure 2: Comparison of cosine similarity across layers for diffuse and wave residual connections with 1-sigma interval shading.

Figure 3: Cosine similarity across layers for a mix (+ Full Wave) with 1-sigma interval shading.

## 5.2 Experiments: CV Tasks

In this section, we demonstrate the capacity of the proposed approach for CV tasks. All of our experiments are conducted on the ImageNet dataset [10] including around 1k classes 1.3M images in the training dataset and 50k images in the validation set. Also, we choose 12-layer DeiT [39] as our backbone. The implementation details is given in Appendix A.2.1.

### 5.2.1 Results of ImageNet Classification

Table 3 summarizes our experimental results. The two rows labeled "Diffusion" in the residual column are baseline scores taken from [29]. The other rows with "+ Wave" are our proposed extensions. These results demonstrate the effectiveness and generality of Wavy Transformer: with almost ignorable minimal additional parameters, it also consistently improves baseline models in CV tasks.

Table 3: Experimental evaluation of Wavy Transformer block plugged into DeiT [39]. The number inside the ($\uparrow \cdot$) represents the performance gain compared with the model without Wavy Transformer block. The two rows labeled "Diffusion" are baseline results taken from [29].

| Method | Residual | Input Size | #Layer | #Param | Top-1 Acc (%) |
|---|---|---|---|---|---|
| DeiT-Ti | Diffusion | 224 | 12 | 5.7M | 72.17 |
| DeiT-Ti | Diffusion + Full Wave | 224 | 12 | 5.7M | 72.33 ($\uparrow 0.16$) |
| DeiT-Ti | Diffusion + Light Wave | 224 | 12 | 5.7M | **73.09** ($\uparrow 0.92$) |
| DeiT-Ti + FeatScale | Diffusion | 224 | 12 | 5.7M | 72.35 |
| DeiT-Ti + FeatScale | Diffusion + Full Wave | 224 | 12 | 5.7M | 72.62 ($\uparrow 0.26$) |

**Quantifying Over-Smoothing.** Beyond cosine similarity, we quantify the spectral gap of the attention matrix and node-feature variance. Wave dynamics show a smaller gap and larger variance on CV tasks (Table 4); definitions and additional results on trends of cosine similarity are in Appendix A.2.2.

Table 4: Over-smoothing diagnostics on CV tasks. Values are calculated over validation images. Spectral gap is $1 - |\lambda_2(\mathbf{A})|$ for the attention matrix. Lower gap ($\downarrow$) and higher node ($\uparrow$) indicate mitigated over-smoothing and Inter-class variance is reported only as an auxiliary separability measure.

| Dynamics | Spectral gap ($1 - |\lambda_2|$) | Node-feature var. | Inter-class var. |
|---|---|---|---|
| Diffusion | $0.836 \pm 0.00345$ | $2.480 \pm 0.0778$ | 0.195 |
| + Full Wave | $\mathbf{0.629 \pm 0.00887}$ | $\mathbf{2.609 \pm 0.0902}$ | 0.211 |
| + Light Wave | $0.730 \pm 0.00842$ | $2.109 \pm 0.0696$ | **0.308** |

### 5.2.2 Scaling to Deeper Models

We additionally evaluate CaiT-XXS-24 on ImageNet-1K under the same training recipe as the baseline [40]. The wave residual improves Top-1 accuracy by 1.0 point. The 0.9M parameter reduction arises solely from omitting extra class-attention block; all other settings remain unchanged (Table 5).

Table 5: ImageNet-1K (CaiT-XXS-24). Numbers in $(\uparrow \cdot)$ indicate Top-1 gain over the diffusion-only baseline. The training recipe and input resolution are identical to the baseline [40].

| Method | Residual | Input Size | #Layer | #Param | Top-1 Acc (%) |
|---|---|---|---|---|---|
| CaiT-XXS-24 | Diffusion | 224 | 24 | 12.0M | 77.6 |
| Wavy Transformer | Diffuse + Full Wave | 224 | 24 | 11.1M | **78.6** $(\uparrow 1.0)$ |

### 5.3 Experiments: Sparse-Graph Tasks

We further evaluate on OGBN-Arxiv and OGBN-Proteins [23] using DIFFormer [44] under the authors' training protocol and hyperparameters. Diffusion denotes vanilla DIFFormer; + Light Wave denotes addition of our momentum-only wave residual. Table 6 highlights representative depths. Complete results are provided in Appendix A.3.2. + Light Wave substantially mitigates depth collapse on OGBN-Arxiv and delivers a consistent ROC-AUC gain on OGBN-Proteins.

Table 6: OGB highlights (mean $\pm$ standard deviation over 3 seeds). Arxiv reports accuracy and Proteins reports ROC-AUC; all values are percentages. $\Delta$ denotes absolute improvement.

| Dataset | Metric $\uparrow$ | #Layer | Diffusion | + Light Wave | $\Delta$ |
|---|---|---|---|---|---|
| OGBN-Arxiv | Acc | 7 | $24.44 \pm 4.51$ | $\mathbf{66.73 \pm 0.33}$ | +42.29 |
| OGBN-Proteins | ROC-AUC | 5 | $69.42 \pm 2.31$ | $\mathbf{80.14 \pm 0.67}$ | +10.72 |

### 5.4 Computational Cost

For cost-sensitive settings, we adopt the momentum-only variant (Light Wave), which preserves the second-order residual with negligible overhead. On both BERT and DeiT-Tiny, throughput and memory stay within a few percent of the diffusion baseline (Table 7); complete cost tables, including the Full Wave variant, are provided in Appendix A.4.

Table 7: Computational efficiency of each residual connections on 4 Nvidia V100.

| Model | Variant | Inference | Training | Peak GPU Mem |
|---|---|---|---|---|
| BERT | Diffusion | 101.6 | 415.6 | 18.31 |
| BERT | Light Wave | **101.3** | **436.2** | **18.69** |
| DeiT-Tiny | Diffusion | 2631.1 | 618.6 | 8.25 |
| DeiT-Tiny | Light Wave | **2644.2** | **617.6** | **9.14** |

## 6 Conclusions

In this paper, we investigated the dynamics embedded in attention layers from the viewpoint of graph neural diffusion and introduced Wavy Transformer. We first established an equivalence between the intrinsic dynamics driven by attention layers and a graph neural diffusion on a complete graph. From this perspective, the over-smoothing behavior can be understood as the dissipative nature of diffusion. Based on this insight, we proposed Wavy Transformer, which can be seamlessly integrated into standard diffusive transformer architecture. Our extensive experiments on NLP, CV, and sparse-graph tasks showed that it can capture feature dynamics that is qualitatively different from those driven by conventional transformers, and that integrating it into existing models enhances performance without increasing the number of training parameters or requiring additional hyperparameter tuning. We believe this work represents a first step toward a physics-inspired design of transformer architecture.

## Acknowledgement

A part of numerical simulations were conducted using Earth Simulator at JAMSTEC.

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

# A  Experimental Details and Additional Results

## A.1  NLP Tasks

### A.1.1  Implementation Details

Following [11], we construct a English corpus from Wikipedia [11] and BooksCorpus [50], splitting it into 95 % training and 5 % validation data. We tokenize using a WordPiece-like tokenizer (30k vocabulary). The WordPiece embedding [46] and the dictionary containing 30,000 tokens [11] are used in our paper. To pre-process text, we use the special token [CLS] as the first token of each sequence and [SEP] to separate sentences in a sequence. Each sequence is up to 128 tokens, with masked language modeling (MLM) applied to 15 % of tokens and next sentence prediction as a binary classification of segment continuity.

We implement three types of models with different residual connections such as diffuse, wave, and mix in a 24-layer BERT with hidden size 256 and 4 attention heads based on Post-LN Wavy Transformer blocks. As for feed-forward layer, we set the intermediate size to 1024. For both wave and mixed residual connections, we set the time interval to $\tau = 0.5$ unless stated otherwise. In this experiment, the mixing coefficient was defined as $\lambda = \text{sigmoid}(\theta)$, with $\theta$ being a single scalar parameter initialized to 0; consequently, the same $\lambda$ is shared across all feature dimensions. For the mixing strategy with Full Wave, we adopt the velocity-based variant: $\mathbf{Y}^{l+1} = \lambda\left[\tau(\mathbf{A}-\mathbf{I})\mathbf{X}^l + \mathbf{Y}^l\right] + (1-\lambda)(\mathbf{A}-\mathbf{I})\mathbf{X}^l$. As a optimizer, we use AdamW [27] with learning rate $5.0 \times 10^{-5}$, warm-up for first 10 % of steps, then linear decay to step 10k. Also, we evaluate every 5k steps on the validation set, reporting PPL and MLM accuracy.

The hyper-parameters of various downstream tasks are shown in Table 8.

Table 8: Hyper-parameters for different downstream tasks.

|  | GLUE | SQuAD |
|---|---|---|
| Batch size | 32 | 16 |
| Weight decay | [0.1, 0.01] | [0.1, 0.01] |
| Warmup proportion | 0.1 | 0.1 |
| Learning rate decay | linear | linear |
| Training epochs | 3 | 3 |
| Learning rate | 0.00005 | 0.00005 |

### A.1.2  MLM Convergence

As shown in Table 9, adding wave terms yields a mild speedup—Mix (+Full) leads early (e.g., +2.6 at 40k and +1.7 at 60k vs. Diffusion)—while Full Wave largely tracks Diffusion thereafter. In short, wave tends to accelerate convergence slightly, but the residual choice does not substantially alter the overall convergence behavior.

Table 9: MLM accuracy vs. steps.

| Model | 0 | 20k | 40k | 60k | 80k | 100k |
|---|---|---|---|---|---|---|
| Diffusion | 0.0 | 17.38 | 32.99 | 40.88 | 43.51 | 44.39 |
| Full Wave | 0.0 | 18.40 | 33.48 | 40.80 | 43.67 | 44.52 |
| Mix (+Full) | 0.0 | 19.40 | 35.56 | 42.60 | 44.77 | 45.56 |

## A.2  CV Tasks

### A.2.1  Implementation Details

To demonstrate that our proposed approach are beneficial to CV tasks, we choose 12-layer DeiT [39] as our backbone. Also, for evaluating its consistency with the other remedy to avoid over-smoothing, we combine our approach with existing technique such as FeatScale [43]. When we use 12-layer

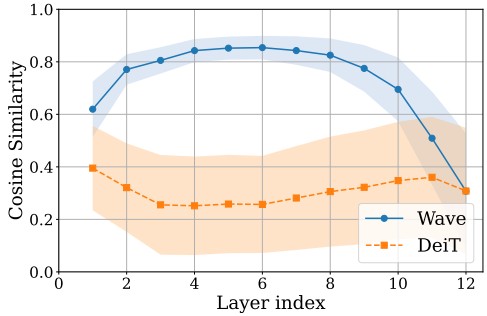 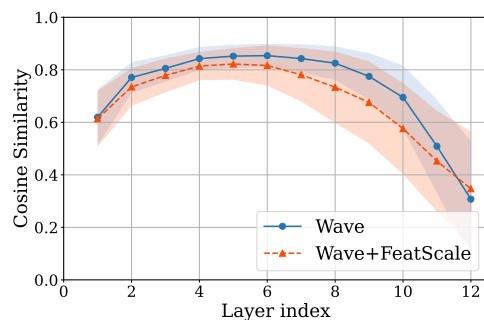

(a) Comparison between cosine similarities with 1-sigma interval shading of the conventional DeiT and the DeiT integrated with Wavy Transformer based on Full Wave variant.

(b) Comparison between cosine similarities with 1-sigma interval shading of the DeiT integrated with Wavy Transformer based on Full Wave variant with and without FeatScale.

Figure 4: Comparisons of cosine similarities with 1-sigma interval shading for several DeiT-based models.

DeiT, we follow the same training recipe, hyper-parameters, and data augmentation with [39, 43]. We implemented DeiT based on Pre-LN Wavy Transformer blocks considering that original DeiT is based on Pre-LN Transformer block. We set $\tau = 0.5$, and we select $\mathbf{X}^{l+1} = \boldsymbol{\lambda}\,\mathbf{X}^{l+1}_{\text{wave}} + (1 - \boldsymbol{\lambda})\,\mathbf{X}^{l+1}_{\text{diffuse}}$, as the mixing strategy with Full Wave and initialize the gating parameter $\boldsymbol{\lambda}$ as an all-zero vector whose length equals the dimension of the hidden states in the mix variant.

### A.2.2   Analysis of Over-smoothing for DeiT-based models

Figure 4 plots similarity curves for the standard DeiT, DeiT integrated with Wavy Transformer based on Full Wave variant, and Wavy Transformer + FeatScale variant with 1-sigma interval shading. The results clearly show how Wavy Transformer reshapes feature dynamics relative to the conventional DeiT. Integrating FeatScale slightly reduces over-smoothing overall, although its final-layer cosine similarity remains marginally higher than that of Wavy Transformer alone.

### A.2.3   Computation of spectral gap and variance metrics

We complement Figure 4 by detailing how we compute the three diagnostics used in the main text (Table 4). Results for spectral gap, node-feature variance, and inter-class variance are reported in the main paper; the definitions here are for reproducibility.

**Spectral Gap**   For each image, we average multi-head attention over heads to obtain a row-/right-stochastic matrix $\mathbf{A} \in \mathbb{R}^{n \times n}$ of the final layer. Let $\{\lambda_i\}_{i=1}^{n}$ be the eigenvalues of $\mathbf{A}$ ordered by magnitude. The per-sample spectral gap is

$$\delta \ = \ 1 - |\lambda_2(\mathbf{A})|,$$

computed by taking the second-largest eigenvalue magnitude of $\mathbf{A}$ (via an eigen decomposition of the head-averaged $\mathbf{A}$), then averaged over the validation set. Smaller $\delta$ indicates weaker mixing and thus less over-smoothing.

**Node-Feature Variance**   Let $\mathbf{X} \in \mathbb{R}^{n \times d}$ be the token (patch) embeddings from a designated layer at evaluation time. We compute the variance across tokens after the final LN for each channel and then average across channels:

$$\text{NodeVar} \ = \ \frac{1}{d} \sum_{j=1}^{d} \text{Var}_{i=1..n}\big(\mathbf{X}_{ij}\big).$$

We report the mean of this scalar over validation images.

**Inter-Class Variance**   Let $\mathcal{K}$ be the set of classes. Tokens inherit their image labels; for each $k \in \mathcal{K}$, define the class centroid $\mu_k$ as the mean token embedding. We then compute the variance across class centroids and average over channels:

$$\text{InterClassVar} = \frac{1}{d} \sum_{j=1}^{d} \text{Var}_{k \in \mathcal{K}}(\mu_{k,j}).$$

This measures feature separability (often correlating with accuracy) and is *not* used as an over-smoothing diagnostic.

### A.2.4   Ablation of FFN for Velocity

Table 10 shows the result of ablation of FFN for velocity. In the Full Wave setting, the FFN for velocity yields a modest but consistent gain (+0.53 Top-1), indicating its effectiveness. However, considering the added computational cost, the momentum-only Light Wave is the more practical choice for cost-sensitive use (see Appendix A.4).

Table 10: Ablation: removing Velocity-FFN (DeiT-Tiny, 150 epochs).

| Configuration | Top-1 (%) | $\Delta$ |
|---|---|---|
| Full Wave (diff+wave+FFN+LN) | 68.92 | – |
| Without FFN for velocity | 68.39 | –0.53 |

## A.3   Sparse-Graph Tasks

### A.3.1   Implimentation Details

We use the momentum-only Light Wave residual for all sparse-graph experiments; the mixing coefficient $\lambda$ is set as a scalar parameter initialized to $0$ at the start of training and optimized jointly with the network. All other settings follow the DIFFormer protocol unless otherwise noted. Also, we adopt the DIFFormer [44] definition of diffusion mixing $\tau$. This is corresponding to the time interval $\tau^2$ in Wavy Transformer. Thus, throughout we control both with a single knob and report $\tau$; see Appendix B for more details.

### A.3.2   OGB: full results and $\tau$-robustness

We report complete results across depths and the coefficient $\tau$. At larger $\tau$, diffusive stacks exhibit collapse at depth, whereas the wave residual mitigates it (Table 11).

Table 11: OGB results (3 seeds). Arxiv: accuracy (%); Proteins: ROC-AUC (%).

| Dataset | Depth | Diffusion | + Light Wave | $\Delta$ |
|---|---|---|---|---|
| Arxiv | 3 | **68.77 $\pm$ 0.28** | 66.68 $\pm$ 0.66 | -2.09 |
| Arxiv | 5 | 61.72 $\pm$ 1.11 | **67.06 $\pm$ 0.03** | +5.34 |
| Arxiv | 7 | 24.44 $\pm$ 4.51 | **66.73 $\pm$ 0.33** | +42.29 |
| Proteins | 3 | 79.26 $\pm$ 0.40 | **79.58 $\pm$ 0.27** | +0.32 |
| Proteins | 5 | 69.42 $\pm$ 2.31 | **80.14 $\pm$ 0.67** | +10.72 |

### A.3.3   Citation Graphs Benchmark

In Tables 12 and 13, we report full results under two mixing regimes, $\tau$=0.2 (moderate) and $\tau$=0.5 (strong neighbor mixing), using the same protocol and hyperparameters; numbers are mean and standard deviation across 5 seeds. Across datasets, + Light Wave matches Diffusion at shallow depth and markedly stabilizes deeper stacks: at $\tau$=0.2 it reverses collapse on Cora/PubMed at 20 layers, with a single significant drop at PubMed 16 layers; at $\tau$=0.5 the diffusive baseline collapses broadly, while + Light Wave mitigates collapse on Cora 12 layers and PubMed 12/20 layers (both variants collapse on Cora 20 layers and Citeseer 10 layers). Significance symbols follow the captions (†: no significant change; ▼: significant drop; ‡: both models collapse).

Table 12: Citation graphs at $\tau = 0.2$ (5 seeds). †: paired $t$-test $p > 0.05$; ▼: only significant drop.

| Dataset | Depth | Diffusion | + Light Wave | Δ |
|---|---|---|---|---|
| Cora | 4 | **80.02 ± 0.29** | 78.54 ± 1.14 | −1.48[†] |
| Cora | 16 | 83.28 ± 0.76 | **84.22 ± 0.94** | +0.94 |
| Cora | 20 | 39.92 ± 5.14 | **85.18 ± 0.48** | +45.26 |
| Citeseer | 4 | **72.04 ± 0.25** | 71.74 ± 0.27 | −0.30[†] |
| Citeseer | 8 | 62.86 ± 17.73 | **69.32 ± 13.26** | +6.46 |
| Citeseer | 10 | 34.76 ± 1.44 | **34.92 ± 2.49** | +0.16 |
| PubMed | 4 | **75.74 ± 0.73** | 74.76 ± 1.15 | −0.98[†] |
| PubMed | 16 | **80.46 ± 0.84** | 78.86 ± 0.62 | −1.60▼ |
| PubMed | 20 | 58.68 ± 2.73 | **79.22 ± 0.93** | +20.54 |

Table 13: Citation graphs at $\tau = 0.5$. ‡: both models collapse.

| Dataset | Depth | Diffusive | + Light Wave | Δ |
|---|---|---|---|---|
| Cora | 12 | 29.40 ± 0.00 | **60.38 ± 24.7** | +31.0 |
| Cora | 20 | 29.40 ± 0.00 | 29.00 ± 0.00 | ±0.0[‡] |
| Citeseer | 10 | 22.94 ± 0.31 | 22.84 ± 2.17 | −0.10[‡] |
| PubMed | 12 | 49.16 ± 9.70 | **69.74 ± 15.7** | +20.6 |
| PubMed | 20 | 29.70 ± 0.00 | **40.10 ± 0.84** | +10.4 |

### A.3.4 Cosine similarity across depth

To quantify over-smoothing for sparse-graph benchmarks, we track the pairwise cosine similarity between node embeddings after each layer in a 20-layer stack on Cora with $\tau = 0.2$. We report means over 5 seeds. Higher values (approaching 1.0) indicate stronger representation collapse within a layer. As shown in Table 14, the diffusion-only residual rapidly saturates to $\approx 1.00$, whereas the wave residual maintains substantially lower similarity across depth, preserving representation diversity.

Table 14: Layer-wise pairwise cosine similarity across node features on Cora (20 layers; 5 seeds). Higher means more over-smoothing.

| Model | Layer 1 | Layer 4 | Layer 8 | Layer 12 | Layer 16 | Layer 20 |
|---|---|---|---|---|---|---|
| Diffusion | 0.99 | 1.00 | 1.00 | 1.00 | 1.00 | 1.00 |
| Wave | 0.31 | 0.058 | 0.087 | 0.13 | 0.23 | 0.38 |

### A.4 Full Results of Computational Cost

In Tables 15 and 16, we report full runtime and memory measurements for three residual variants under the same training scripts and hyperparameters as the accuracy experiments. Metrics include inference throughput (k tokens/s for BERT; img/s for DeiT-Tiny), training iteration time (ms/iter), and peak GPU memory (GB), all on 4×V100. Only the residual dynamics are changed, isolating their compute impact.

### A.4.1 Throughput–Accuracy Trade-off Comparison for Full Wave Branch

Table 17 compares the inference throughput (images/s) of various transformer variants, measured on the ImageNet validation set using V100 GPUs and a batch size of 256. Introducing wave dynamics (Diffuse+Wave or Wave alone) reduces throughput by roughly 50% from about 2 600 to 1 200–1 300 images per second which may pose a limitation in real-time applications. However, this slowdown must be balanced against the gains in representational power and accuracy that wave-enhanced models provide. A simple mitigation is to insert wavy blocks in only a subset of layers to recover part of the lost throughput. Concretely, replacing only the last six blocks with wavy blocks (Wave (6))

Table 15: BERT: runtime and memory on 4×V100.

| Variant | Inference (k tokens/s) | Training (ms/iter) | Peak GPU (GB) |
|---|---|---|---|
| Diffusion | 101.6 | 415.6 | 18.31 |
| + Full Wave | 63.3 | 1031 | 28.71 |
| + Light Wave | 101.3 | 436.2 | 18.69 |

Table 16: DeiT-Tiny: runtime and memory on 4×V100.

| Variant | Inference (img/s) | Training (ms/iter) | Peak GPU (GB) |
|---|---|---|---|
| Diffusion | 2631.1 | 618.6 | 8.25 |
| + Full Wave | 1312.2 | 1023.4 | 20.31 |
| + Light Wave | 2644.2 | 617.6 | 9.14 |

increases throughput by about 33 % relative to the full Diffuse+Wave with FeatScale variant (1 649 img/s vs. 1 241 img/s) while sacrificing 0.18 pt in Top-1 accuracy (72.44 % vs. 72.62 %). In this experiment, the velocity tensor was initialized to zero at the first wavy block.

Table 17: Inference throughput and accuracy on ImageNet. Wave (6) indicates that wavy blocks are inserted only in the final six layers.

| Model | Residual Connections | Throughput | Top-1 (%) |
|---|---|---|---|
| DeiT-Ti [39] | Diffusion | **2632.1** | 72.17 |
| DeiT-Ti + FeatScale [43] | Diffusion | 2483.2 | 72.35 |
| DeiT-Ti | Diffusion+ Full Wave | 1266.0 | 72.33 |
| DeiT-Ti + FeatScale | Diffusion+ Full Wave | 1241.4 | **72.62** |
| DeiT-Ti | Full Wave | 1312.2 | – |
| DeiT-Ti + FeatScale | Diffuse+ Full Wave (6) | 1649.3 | 72.44 |

## A.5 Wavy Transformer under causal attention

For supplemental information, we test the Light Wave residual under strictly causal attention using `nanoGPT` [25]. We use the OpenWebText dataset and initialize from an OpenAI GPT-2 checkpoint, keeping the standard GPT causal mask and training loop unchanged. During fine-tuning, we use a batch size of 1 with gradient accumulation over 32 steps; training runs for 50 iterations with a learning rate of $3 \times 10^{-5}$ and learning-rate decay enabled. Under this setup, the best validation loss, averaged over three seeds, was 3.0178 for the Light Wave mix (scalar $\lambda$ initialized to 0) versus 3.0196 for the diffusion baseline—an absolute gap of 0.0018 ($\approx 0.06\%$); while the Light Wave mix is slightly better, we do not observe a meaningful difference in validation loss. A more comprehensive assessment under causal masking requires substantially longer training; we therefore view this as worthwhile future work, related to potential over-smoothing in causal-attention Transformers. From a diffusive-dynamics perspective, the attention matrix acts as a diffusion-coefficient matrix; under causal masking it becomes strictly unidirectional along the sequence, so causal attention can be viewed as diffusion on a directed complete graph with unidirectional edges.

## B Implementation Details of Light Wave

As shown in Section 3.2, if we directly discretize Eq. (9) in time, rather than bypassing it through Eq. (10), we obtain

$$\mathbf{X}^{l+1} = \tau^2 \mathbf{A} \mathbf{X}^l + (1 - \tau^2)\mathbf{X}^l + (\mathbf{X}^l - \mathbf{X}^{l-1}). \tag{16}$$

This equation can be seen as adding the momentum term $(\mathbf{X}^l - \mathbf{X}^{l-1})$ into the usual diffusive attention without considering the coefficient $\tau^2$. Also, if we consider to blend the outputs of the diffusive attention layer and the proposed wavy attention layer:

$$\mathbf{X}^{l+1} = \lambda \, \mathbf{X}_{\text{wave}}^{l+1} + (1 - \lambda) \, \mathbf{X}_{\text{diffuse}}^{l+1},$$

where the mixing vector $\boldsymbol{\lambda} \in [0, 1]^d$ is defined as $\boldsymbol{\lambda} = \text{sigmoid}(\boldsymbol{\theta})$, with $\boldsymbol{\theta} \in \mathbb{R}^d$ a trainable parameter vector.

**Decoupling $\tau$ Across Diffusion and Wave.** Let $\tau_{\mathrm{d}}$ and $\tau_{\mathrm{w}}$ be the (possibly different) time intervals used in the diffusive and wavy paths, respectively. Substituting

$$\mathbf{X}_{\text{diffuse}}^{l+1} = (1 - \tau_{\mathrm{d}}) \, \mathbf{X}^l + \tau_{\mathrm{d}} \, \mathbf{A}\mathbf{X}^l, \tag{17}$$

$$\mathbf{X}_{\text{wave}}^{l+1} = (1 - \tau_{\mathrm{w}}^2) \, \mathbf{X}^l + \tau_{\mathrm{w}}^2 \, \mathbf{A}\mathbf{X}^l + \left( \mathbf{X}^l - \mathbf{X}^{l-1} \right), \tag{18}$$

into the blend gives the elementwise form

$$\mathbf{X}^{l+1} \;=\; (1 - \bar{\rho}) \, \mathbf{X}^l + \bar{\rho} \, \mathbf{A}\mathbf{X}^l \;+\; \boldsymbol{\lambda} \left( \mathbf{X}^l - \mathbf{X}^{l-1} \right), \qquad \bar{\rho} := \boldsymbol{\lambda} \tau_{\mathrm{w}}^2 + (1 - \boldsymbol{\lambda}) \tau_{\mathrm{d}}. \tag{19}$$

Equation (19) shows that it is *not necessary* to strictly match $\tau$ between the two paths: only the *effective* diffusion mix $\bar{\rho}$ and the momentum weight $\boldsymbol{\lambda}$ govern the update. Equivalently, by reparameterizing with a single knob $\tau := \bar{\rho} \in [0, 1]$, the update reduces to the Light Wave form used in our experiments.

## C  Energy Evolution in Diffusion and Wave Systems

We provide the fundamental theorems [15, 14, 36] that characterize behavior of Eqs.(3) and (4): energy dissipation in the diffusion equation and energy conservation in the wave equation.

**Theorem 1** *Consider a solution to the diffusion equation. Then, the following holds:*

$$\frac{d}{dt} \int (x(s,t))^2 ds = -2 \int |\nabla x(s,t)|^2 ds \leq 0,$$

*which implies that the norm of the solution is monotonically non-increasing in time.*

**Theorem 2** *Define the energy functional for $x = x(s,t)$ which is a solution to the wave equation*

$$E(t) = \frac{1}{2} \int \left[ \left( \frac{\partial x(s,t)}{\partial t} \right)^2 + |\nabla x(s,t)|^2 \right] ds.$$

*Then, under suitable smoothness and boundary conditions, we have:*

$$\frac{d}{dt} E(t) = 0.$$

*so $E(t)$ is constant in time.*

## D  Appendices of Difference Between Pre-LN and Post-LN

In this section, we discuss the analytical and implemental difference between Pre-LN and Post-LN. This section provides a concise review of the differences between Pre-LN and Post-LN, presents a physical interpretation of Pre-LN attention as a diffusion–reaction process, and details of the computational flow of Pre-LN and Post-LN Wavy Transformers.

### D.1  Pre-LN vs. Post-LN

Figure 5 contrasts the computational flows of the Post-LN and Pre-LN Wavy Transformers. The critical difference is whether LN is applied inside or outside the residual connection. For reference, Table 18 compares the core computational flows of standard Post-LN and Pre-LN transformer blocks; the table was compiled with reference to [47]. In a Post-LN Transformer (the original Vaswani et al. design [41]), each attention or FFN is applied first, and then LN is performed on the sum of the sub-layer output and the residual shortcut. This keeps the summed output "clean", but it is known to slow or destabilize training [47]. In a Pre-LN Transformer, each block first normalizes its input with LN and then applies the sub-layer (self-attention or feed-forward) before adding the residual connection, in contrast to the original "Post-LN" design that normalizes after the addition. Placing LN at the block's entrance ensures every sub-layer receives standardized inputs. In the case of Pre-LN, it is reported that the gradient are well-behaved at initialization [47].

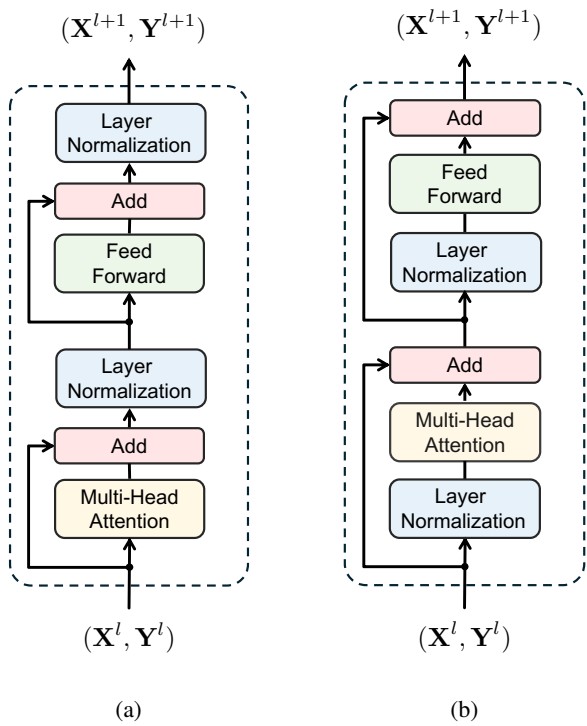

| | |
|---|---|
| (a) | (b) |

Figure 5: (a) Post-LN Wavy Transformer layer; (b) Pre-LN Wavy Transformer layer.

Table 18: Post-LN Transformer vs. Pre-LN Transformer

| Post-LN Transformer | Pre-LN Transformer |
|---|---|
| $\mathbf{X}^{l,1} = \text{Attn}(\mathbf{X}^l)$ | $\mathbf{X}^{l,1} = \text{LayerNorm}(\mathbf{X}^l)$ |
| $\mathbf{X}^{l,2} = \mathbf{X}^{l,1} + \mathbf{X}^l$ | $\mathbf{X}^{l,2} = \text{Attn}(\mathbf{X}^{l,1})$ |
| $\mathbf{X}^{l,3} = \text{LayerNorm}(\mathbf{X}^{l,2})$ | $\mathbf{X}^{l,3} = \mathbf{X}^l + \mathbf{X}^{l,2}$ |
| $\mathbf{X}^{l,4} = \text{FFN}(\mathbf{X}^{l,3})$ | $\mathbf{X}^{l,4} = \text{LayerNorm}(\mathbf{X}^{l,3})$ |
| $\mathbf{X}^{l,5} = \mathbf{X}^{l,3} + \mathbf{X}^{l,4}$ | $\mathbf{X}^{l,5} = \text{FFN}(\mathbf{X}^{l,4})$ |
| $\mathbf{X}^{l+1} = \text{LayerNorm}(\mathbf{X}^{l,5})$ | $\mathbf{X}^{l+1} = \mathbf{X}^{l,5} + \mathbf{X}^{l,3}$ |
| | $\mathbf{X}_{\text{Final}} \leftarrow \text{LayerNorm}(\mathbf{X}^{l+1})$ |

### D.2 Pre-LN vs. Post-LN Implementations in Wavy Transformer

For considering Wavy Transformer with Full Wave setting, the difference between Pre-LN and Post-LN [47] is not trivial. Therefore, this section examines the implementation differences between Pre-LN and Post-LN in the context of Wavy Transformer. Building on the Pre-LN and Post-LN baseline implementations shown Table 18, we construct Post-LN and Pre-LN versions of Wavy Transformer, whose details are summarized in Table 19. Conceptually, Wavy Transformer replaces the conventional residual update based on diffusive dynamics with one governed by wavy dynamics.

When the wavy residual update is combined with the conventional (diffusive) attention, the choice between Pre-LN and Post-LN becomes especially significant: a mismatch in normalization placement can disrupt the balance between the wavy and diffusive paths and degrade accuracy. Consequently, the LN configuration of wavy branch must be aligned with that of the diffusive branch to achieve the intended performance improvements. In our NLP experiments, we adopted Post-LN Wavy Transformer. Conversely, because DeiT [39] is built on a Pre-LN backbone, we used Pre-LN Wavy Transformer for all DeiT-based CV experiments.

Table 19: Post-LN Wavy-Transformer vs. Pre-LN Wavy-Transformer

| Post-LN Wavy-Transformer | Pre-LN Wavy-Transformer |
|---|---|
| $\mathbf{X}^{l,1} = \mathrm{Attn}\big(\mathbf{X}^l\big)$ | $\mathbf{X}^{l,1}, \mathbf{Y}^{l,1} = \mathrm{LayerNorm}\big(\mathbf{X}^l, \mathbf{Y}^l\big)$ |
| $\mathbf{Y}^{l,1} = \tau\big(\mathbf{X}^{l,1} - \mathbf{X}^l\big) + \mathbf{Y}^l$ | $\mathbf{X}^{l,2} = \mathrm{Attn}\big(\mathbf{X}^{l,1}\big)$ |
| $\mathbf{X}^{l,1} = \tau\mathbf{Y}^{l,1} + \mathbf{X}^l$ | $\mathbf{Y}^{l,2} = \tau\big(\mathbf{X}^{l,2} - \mathbf{X}^l\big) + \mathbf{Y}^l$ |
| $\mathbf{X}^{l,2}, \mathbf{Y}^{l,2} = \mathrm{LayerNorm}\big(\mathbf{X}^{l,1}, \mathbf{Y}^{l,1}\big)$ | $\mathbf{X}^{l,3} = \tau\mathbf{Y}^{l,2} + \mathbf{X}^l$ |
| $\mathbf{X}^{l,3}, \mathbf{Y}^{l,2} = \mathrm{FFN}\big(\mathbf{X}^{l,2}, \mathbf{Y}^{l,2}\big)$ | $\mathbf{X}^{l,4}, \mathbf{Y}^{l,3} = \mathrm{LayerNorm}\big(\mathbf{X}^{l,3}, \mathbf{Y}^{l,2}\big)$ |
| $\mathbf{X}^{l,4} = \mathbf{X}^{l,2} + \mathbf{X}^{l,3}, \mathbf{Y}^{l,4} = \mathbf{Y}^{l,2} + \mathbf{Y}^{l,3}$ | $\mathbf{X}^{l,5}, \mathbf{Y}^{l,4} = \mathrm{FFN}\big(\mathbf{X}^{l,4}, \mathbf{Y}^{l,3}\big)$ |
| $\mathbf{X}^{l+1}, \mathbf{Y}^{l+1} = \mathrm{LayerNorm}\big(\mathbf{X}^{l,5}, \mathbf{Y}^{l,5}\big)$ | $\mathbf{X}^{l+1} = \mathbf{X}^{l,5} + \mathbf{X}^{l,3}, \mathbf{Y}^{l+1} = \mathbf{Y}^{l,4} + \mathbf{Y}^{l,2}$ |
| | $\mathbf{X}_{\mathrm{Final}}, \mathbf{Y}_{\mathrm{Final}} \leftarrow \mathrm{LayerNorm}\big(\mathbf{X}^{l+1}, \mathbf{Y}^{l+1}\big)$ |

### D.3 Attention with Pre-LN as graph neural diffusion with reaction term

In Section 3, it is shown that the attention with Post-LN can be straightforwardly considered as a graph neural diffusion. In addition to that, we show that attention with Pre-LN also can be regarded as a graph neural diffusion. First of all, attention with Pre-LN can be formalized as follow:

$$\mathbf{X}^{l+1} = \mathrm{Attn}\big(\mathrm{LN}(\mathbf{X}^l)\big) + \mathbf{X}^l. \tag{20}$$

We compactly write LN transformation in matrix-form as

$$\mathrm{LN}(\mathbf{X}) = \tilde{\mathbf{X}} = \mathbf{\Sigma}^{-1}\mathbf{X}\mathbf{S} + \mathbf{M}, \tag{21}$$

where $\mathbf{\Sigma} := \mathrm{diag}\big(\sigma_i\big) \in \mathbb{R}^{n \times n}$, $\mathbf{S} := \mathrm{diag}\big(\gamma_j\big) \in \mathbb{R}^{d \times d}$. Also, $\mathbf{M} := -\big(\frac{\boldsymbol{\mu}}{\boldsymbol{\sigma}}\big)\boldsymbol{\gamma}^\top + \mathbf{1}_n\boldsymbol{\beta}^\top \in \mathbb{R}^{n \times d}$, where $\boldsymbol{\mu} = (\mu_1, \ldots, \mu_n)^\top$, $\boldsymbol{\sigma} = (\sigma_1, \ldots, \sigma_n)^\top$, $\boldsymbol{\gamma} = (\gamma_1, \ldots, \gamma_d)^\top$, and $\boldsymbol{\beta} = (\beta_1, \ldots, \beta_d)^\top$. Using this representation, we can obtain

$$\mathrm{Attn}(\tilde{\mathbf{X}}) = \mathrm{softmax}\left(\frac{\tilde{\mathbf{X}}\mathbf{W}_q(\tilde{\mathbf{X}}\mathbf{W}_K)^\top}{\sqrt{d}}\right)\tilde{\mathbf{X}}\mathbf{W}_V \tag{22}$$

$$= \tilde{\mathbf{A}}\mathbf{\Sigma}^{-1}\mathbf{X}\mathbf{S}\mathbf{W}_V + \tilde{\mathbf{M}}\mathbf{W}_V \tag{23}$$

$$= \big(\tilde{\mathbf{A}}\mathbf{\Sigma}^{-1}\mathbf{X}\mathbf{S} + \tilde{\mathbf{M}}\big)\mathbf{W}_V \tag{24}$$

where $\tilde{\mathbf{A}} = \mathrm{softmax}\left(\frac{\tilde{\mathbf{X}}\mathbf{W}_q(\tilde{\mathbf{X}}\mathbf{W}_K)^\top}{\sqrt{d}}\right)$ and $\tilde{\mathbf{M}} = \tilde{\mathbf{A}}\mathbf{M}$. Therefore, attention with Pre-LN described in Eq.(20) can be written as

$$\mathbf{X}^{l+1} = \big(\tilde{\mathbf{A}}\mathbf{\Sigma}^{-1}\mathbf{X}\mathbf{S} + \tilde{\mathbf{M}}\big)\mathbf{W}_V + \mathbf{X}^l. \tag{25}$$

On the other hand, consider the graph neural equation with an external source term and a reaction term as follows

$$\frac{\partial \mathbf{X}}{\partial t} = \big(\tilde{\mathbf{A}} - \mathbf{I}\big)\mathbf{\Sigma}^{-1}\mathbf{X}\mathbf{S} + \tilde{\mathbf{M}} + \mathbf{\Sigma}^{-1}\mathbf{X}\mathbf{S}. \tag{26}$$

For simplicity, we temporarily ignore the feature transformation by $\mathbf{W}_V$. Straightforwardly, considering discretization of this equation with $\tau = 1.0$ recover the update rule of the attention with Pre-LN written as Eq.(25). The continuous counterpart can be considered as the following the diffusion equation with an external source term and reaction term:

$$\frac{\partial x(s,t)}{\partial t} = \underbrace{\mathrm{div}\big(D \nabla x(s,t)\big)}_{\text{diffusion}} + \underbrace{b(u,t)}_{\text{source}} - \underbrace{\kappa(u,t)\, x}_{\text{reaction}}. \tag{27}$$

In some cases, Pre-LN Transformers have been shown to outperform their Post-LN counterparts [47]. Furthermore, in GNNs, incorporating diffusion–reaction dynamics has led to performance gains [6]. Together with the interpretation of Pre-LN attention as a diffusion–reaction process, these findings suggest that the superiority of Pre-LN arises from the implicit dynamics it introduces, which help mitigate over-smoothing and enhance model performance. In this sense, a physical-dynamics perspective offers a promising basis for clarifying transformer behavior.

## E Intuitive Explanation of Over-smoothing

The graph diffusion equation [5] can be rewritten as

$$\frac{\partial \mathbf{X}}{\partial t} = \mathbf{A}\mathbf{X} - \mathbf{X} = \bar{\mathbf{X}} - \mathbf{X}. \tag{28}$$

Here, we define the attention-weighted average of the hidden states as $\bar{\mathbf{X}} = \mathbf{A}\mathbf{X}$, where $\mathbf{A}$ is right-stochastic matrix. This relation clearly explain why transformers suffer from over-smoothing: if some a hidden state $\mathbf{X}$ exceeds its attention-weighted average $\bar{\mathbf{X}}$, it is driven downward, whereas if it falls below $\bar{\mathbf{X}}$, it is pulled upward. Consequently, as the hidden states propagate through many layers, an effect analogous to integrating the diffusion process over a long time, every component $\mathbf{X}$ converges toward the common value characterized by $\bar{\mathbf{X}}$. The farther a token is from that local mean, the stronger the restoring force, so high-variance patterns disappear first. What remains is the slowly changing, globally coherent structure, which is exactly the smoothing effect in the neural graph diffusion. We can summarize as follows. Discretization of Eq. (28) gives:

$$\mathbf{X}_i^{l+1} - \mathbf{X}_i^l = \tau\big(\bar{\mathbf{X}}_i^l - \mathbf{X}_i^l\big), \quad 0 < \tau < 1. \tag{29}$$

By simple case analysis:

$$\mathbf{X}_i^{l+1} - \mathbf{X}_i^l = \tau\big(\bar{\mathbf{X}}_i^l - \mathbf{X}_i^l\big) \begin{cases} < 0, & \mathbf{X}_i^l > \bar{\mathbf{X}}_i^l, \\ = 0, & \mathbf{X}_i^l = \bar{\mathbf{X}}_i^l, \\ > 0, & \mathbf{X}_i^l < \bar{\mathbf{X}}_i^l. \end{cases} \tag{30}$$

Hence each step strictly reduces the deviation from the mean:

$$\big|\mathbf{X}_i^{l+1} - \bar{\mathbf{X}}_i^l\big| = (1 - \tau)\big|\mathbf{X}_i^l - \bar{\mathbf{X}}_i^l\big| < \big|\mathbf{X}_i^l - \bar{\mathbf{X}}_i^l\big|. \tag{31}$$

## F Energy Evaluation in Diffusion and Wave Dynamics on Graphs

In this section, to indicate an intuitive discussion of diffusion and wave dynamics on graphs from an energetic viewpoint, we assume here time-independent attention matrix $\mathbf{A}$ (e.g. $\mathbf{A}_{ij} = \mathbf{A}(\mathbf{X}_i^0, \mathbf{X}_j^0)$ using the initial features) and impose the symmetry condition $\mathbf{A}_{ij} = \mathbf{A}_{ji}$. These simplifying assumptions do not hold exactly in real transformers; they are introduced here solely for analytical simplicity and an intuitive understanding of their hidden-state dynamics. Note that this section presents the discrete-graph analogue of Appendix C and should be compared with it.

**Graph Neural Diffusion as Gradient Flow** Consider the following potential energy:

$$U(\mathbf{X}) = \frac{1}{2} \sum_{i,j} \mathbf{X}_i^\top \big(\mathbf{I} - \mathbf{A}\big)_{ij} \mathbf{X}_j. \tag{32}$$

Here, the gradient of $U(\mathbf{X})$ with respect to $\mathbf{X}$ is

$$\left[\frac{\partial U(\mathbf{X})}{\partial \mathbf{X}}\right]_k = \frac{1}{2}\bigg(\sum_{i,j} \mathbf{I}_{ki}\,(\mathbf{I} - \mathbf{A})_{ij}\,\mathbf{X}_j + \sum_{i,j} \mathbf{X}_i^\top\,(\mathbf{I} - \mathbf{A})_{ij}\,\mathbf{I}_{jk}\bigg) \tag{33}$$

$$= \sum_j (\mathbf{I} - \mathbf{A})_{kj}\,\mathbf{X}_j, \tag{34}$$

where $\mathbf{I}$ is identity matrix and we used the symmetry condition $\mathbf{A}_{ij} = \mathbf{A}_{ji}$. Therefore, the graph neural diffusion can be written as

$$\frac{\partial \mathbf{X}}{\partial t} = -\frac{\partial U(\mathbf{X})}{\partial \mathbf{X}}. \tag{35}$$

This shows that the hidden-state dynamics in transformers, which implicitly rely on graph neural diffusion, can be interpreted as the gradient flow of the potential energy in Eq. (32). A similar discussion appears in [29, 22], and a more general mathematical discussion is given in [16].

Let the node features $\mathbf{X}$ evolve according to the graph neural diffusion in Eq. (7). Using the symmetry of $\mathbf{A}$ and the row-stochastic condition $\sum_j \mathbf{A}_{ij} = 1$, we can rewrite the potential as

$$U(\mathbf{X}) = \frac{1}{4} \sum_{i,j} \mathbf{A}_{ij} \|\mathbf{X}_j - \mathbf{X}_i\|^2, \tag{36}$$

where $\|\mathbf{X}_j - \mathbf{X}_i\|^2 = (\mathbf{X}_j - \mathbf{X}_i)^\top (\mathbf{X}_j - \mathbf{X}_i)$. Taking the time derivative along the gradient flow yields

$$\frac{dU(\mathbf{X})}{dt} = \left\langle \frac{\partial U(\mathbf{X})}{\partial \mathbf{X}}, \frac{\partial \mathbf{X}}{\partial t} \right\rangle = -\left\| \frac{\partial U(\mathbf{X})}{\partial \mathbf{X}} \right\|^2 = -\|(\mathbf{I} - \mathbf{A})\mathbf{X}\|^2 \leq 0, \tag{37}$$

where $\langle \cdot, \cdot \rangle$ denotes the Frobenius inner product, i.e., $\langle \mathbf{Y}, \mathbf{Z} \rangle = \operatorname{tr}(\mathbf{Y}^\top \mathbf{Z})$. If we define the attention-weighted average $\bar{\mathbf{X}} = \mathbf{A}\mathbf{X}$ (cf. Appendix E), then

$$\frac{dU(\mathbf{X})}{dt} = -\|\bar{\mathbf{X}} - \mathbf{X}\|^2 = -\sum_i \|\bar{\mathbf{X}}_i - \mathbf{X}_i\|^2. \tag{38}$$

This demonstrates that the graph neural diffusion decreases the potential energy (36) monotonically, and that $U(\mathbf{X})$ can be interpreted as the sum of squared differences between each feature and its attention-weighted average, implying convergence toward a uniform state. This provides a clear explanation for the over-smoothing observed in transformers from the energetic viewpoint.

**Energy Conservation of Wave Dynamics on Graphs**  In contrast to the graph neural diffusion, wavy dynamics introduced into Wavy Transformer conserve their energy in the same way as continuous dynamics on smooth manifold. To show that, consider the following energy for $(\mathbf{X}, \mathbf{Y})$ governed by Eq. 11:

$$E(\mathbf{X}, \mathbf{Y}) = \frac{1}{2}\sum_i \|\mathbf{Y}_i\|^2 + \frac{1}{2}\sum_{i,j}\mathbf{X}_i^\top (\mathbf{I} - \mathbf{A})_{ij}\mathbf{X}_j. \tag{39}$$

Taking the time derivative and using $\dot{\mathbf{X}}_i = \mathbf{Y}_i$ and $\dot{\mathbf{Y}} = -(\mathbf{I} - \mathbf{A})\mathbf{X}$, we obtain

$$\frac{dE(\mathbf{X})}{dt} = \sum_i \mathbf{Y}_i^\top \dot{\mathbf{Y}}_i + \sum_{i,j}\dot{\mathbf{X}}_i^\top (\mathbf{I} - \mathbf{A})_{ij}\mathbf{X}_j \tag{40}$$

$$= -\sum_i \mathbf{Y}_i^\top \sum_j (\mathbf{I} - \mathbf{A})_{ij}\mathbf{X}_j + \sum_{i,j}\mathbf{Y}_i^\top (\mathbf{I} - \mathbf{A})_{ij}\mathbf{X}_j = 0. \tag{41}$$

Hence, the total energy $E(\mathbf{X}, \mathbf{Y})$ remains constant over time, i.e., throughout the entire forward pass, demonstrating that the wavy dynamics on the graph exactly conserve energy.

