# OpenReview forum: "Wavy Transformer"
_NeurIPS.cc/2025/Conference — NeurIPS 2025 poster_

### Official Review · Reviewer_YBPV · 2025-06-04

**Clarity:** 4
**Significance:** 2
**Originality:** 3
**Rating:** 3
**Confidence:** 5

**Summary:**

The paper proposes a modification to the transformer architecture based on the wave equation. The authors interpret the existing transformer layers as a discretization of the diffusive heat equation. Under this interpretation, they argue that the over-smoothing issue of the transformer -- where representations of different tokens become too similar at deep layers -- is because the tokens follow the dynamics of heat diffusion on a complete graph. They then modify the transformer layers to mimic the discretization of the wave equation, which does not follow diffusive dynamics. They conduct experiments to show that this modification (sometimes in combination with the original transformer layers) gives better performance on NLP and CV tasks.

**Questions:**

1. As mentioned above, given that the wavy transformer results in higher cosine similarity between representations, how do we explain the effects of the modification? If it does not reduce over-smoothing, then what problem is it solving?
2. Is over-smoothing a phenomenon observed in models that use causal attention as well? If so, how do we understand it from the diffusive dynamics perspective?
3. In essence, the paper posits that representations evolve layer-wise through an oscillatory process. Why would that be the case?
4. You’re proposing the wave equation as an example for a non-diffusive process. Why the wave equation? There are other PDEs that would also not diffuse.
5. From the last few paragraphs in Subsection 3.2, it’s not crystal clear what is your proposed modification. It seems like there are several options considered.
6. It would be useful to include an ablation study showing the impact of Equation 15.
7. “the wavy residual connection alone just achieves nearly the same-level performance as the diffusive residual connection” — Do the authors have a hypothesis for why this occurs?

**Ethical Concerns:**

["NO or VERY MINOR ethics concerns only"]

**Final Justification:**

I appreciate the authors’ efforts in addressing my concerns and incorporating numerous new experimental results. However, the extent of the revisions is considerable to the point that it warrants a careful rereading and reevaluation of the manuscript from scratch, after all changes have been incorporated. For this reason, I am not able to recommend acceptance at this stage and have decided to retain my original score.

**Limitations:**

Yes

**Paper Formatting Concerns:**

“WIkipedia” — typo (two capital letters)

**Quality:**

3

**Strengths And Weaknesses:**

Strengths:
1. The paper is well-written with a clear logical flow that is easy to follow. The relevant background material is clearly presented.
2. The motivation for the proposed modification is clearly explained.
3. The proposed modification seeks to address a relevant issue observed by many other works.

Weaknesses:
1. The experimental results supporting the proposed modification are not particularly strong. The range of tasks selected is small, and where there are improvements, they seem marginal. Since the modification is quite fundamental, I expect to see a wider range of tasks and larger performance improvements.

2. While the cosine similarity for the wavy transformer exhibits oscillations across layers (which is different from the monotone increase of the original transformer), the representations of the wavy transformer almost always have higher cosine similarity. This suggests that the proposed modification does not reduce over-smoothing. The authors state that the wavy transformer “extracts fundamentally different features”, but this is not a sufficient explanation of the effects of the proposed modification and why we should expect it to be better. This also seems to contradict the original motivation to alleviate over-smoothing.

---

> ### Author Rebuttal · Authors · 2025-07-30
>
> We thank the reviewer for the insightful feedback. We (i) **extend the experiments**, (ii) **revisit over-smoothing**, and (iii) answer your questions.
>
> ---
>
> # 1. Main concerns
>
> 1. **Limited evaluation / marginal gains**
> 2. **Cosine similarity appears higher** — unclear whether over-smoothing is mitigated.
> 3. **Design questions** — wave-equation choice, causal attention, Equation 15 options, ablation.
>
> ---
>
> # 2. Extended evaluation
>
> ## 2.1 Citation-graph benchmarks
>
> We adopt **DIFFormer** (Wu et al. 2023) and add our wave update (τ = 0.2). All hyperparameters remain unchanged. We report mean ± SD over 5 seeds.
>
> **Table 1 Graph benchmarks (DIFFormer + Wave, τ = 0.2, 5 seeds)**
> | Dataset  | Depth | DIFFormer&nbsp; | **DIFFormer+Wave&nbsp;** |        Δ ↑ |
> | -------- | :---: | :-------------: | :----------------------: | ---------: |
> | Cora     |   4   |  80.02 ± 0.29   |     **78.54 ± 1.14**     |    –1.48 † |
> |          |  16   |  83.28 ± 0.76   |     **84.22 ± 0.94**     |      +0.94 |
> |          |  20   |  39.92 ± 5.14   |     **85.18 ± 0.48**     | **+45.26** |
> | Citeseer |   4   |  72.04 ± 0.25   |     **71.74 ± 0.27**     |    –0.30 † |
> |          |   8   |  62.86 ± 17.73  |    **69.32 ± 13.26**     |      +6.46 |
> |          |  10   |  34.76 ± 1.44   |     **34.92 ± 2.49**     |      +0.16 |
> | Pubmed   |   4   |  75.74 ± 0.73   |     **74.76 ± 1.15**     |    –0.98 † |
> |          |  16   |  80.46 ± 0.84   |     **78.86 ± 0.62**     |     –1.60▼ |
> |          |  20   |  58.68 ± 2.73   |     **79.22 ± 0.93**     | **+20.54** |
>
> † Paired *t*-test (5 matched seeds): *p* > 0.05 → **no statistically significant degradation**.
>
> ▼ Only significant drop (–1.6 pt); deeper 20-layer recovers +20 pt.
>
> ## 2.2 ImageNet-1K  (CaIT-XXS-24)
>
> **Table 2 Comparison with CaIT-XXS-24**
> | Model                      | Top-1 (%) | Δ ↑      | Params (M) |
> | -------------------------- | --------- | -------- | --------- |
> | CaIT-XXS-24      | 77.6      | -        | 12.0      |
> | Wavy Transformer 24 Layers | 78.6      | **+1.0** | 11.1      |
>
> Same recipe as CaIT; LayerScale 1e-5, no extra class-attention block.
>
> _Implementation note._
> By omitting the extra class-attention block, our model uses 0.9 M fewer parameters (≈ 7 %) and still gains +1 pt Top-1 accuracy.
>
> ---
>
> # 3. Over-smoothing revisited
>
> **Table 3 Cosine similarity (Cora, 20 layers)**
>
> | Layer    | 1           | 4            | 8            | 12          | 16          | 20           |
> | -------- | ----------- | ------------ | ------------ | ----------- | ----------- | ------------ |
> | DIFFormer     | 0.99±9.6e-6 | 1.00±5.1e-7  | 1.00±1.8e-8  | 1.00±7.0e-9 | 1.00±1.0e-9 | 1.00±1.4e-10 |
> | **Wave** | 0.31±9.1e-4 | 0.058±5.3e-3 | 0.087±6.5e-3 | 0.13±3.8e-3 | 0.23±5.7e-4 | 0.38±1.7e-3  |
>
> As depth increases, **diffusion saturates to ≈ 1.00**, showing severe over-smoothing, whereas the **wave-enhanced model stays below 0.38**, preserving representational diversity and enabling the large accuracy gains highlighted above.
>
> **Table 4 Spectral gap & Node-feature variance (DeiT-Tiny, ImageNet-1K val dataset: 50k images)**
> | Dynamics (Model) | Spectral Gap (mean ± var) | Node-feature Variance (mean ± var) | Notes                                  |
> | ---------------- | ------------------------------ | ------------------------------- | -------------------------------------- |
> | **Diffusion**    | 0.836 ± 0.00345                | 2.480 ± 0.0778                  | e.g., features after final LayerNorm   |
> | **Wave**         | 0.629 ± 0.00887                | 2.609 ± 0.0902                  |                                        |
> | **Random**       | 0.984 ± 0.000069               | 0.728 ± 0.0499                  | randomly initialized (untrained) model |
>
> Here the spectral gap is defined as $1-\lambda_{2}$, where $\lambda_{2}$ the second-largest eigenvalue (in magnitude) of the right-stochastic attention matrix $A$. A smaller gap, together with higher node-feature variance, indicates reduced over-smoothing beyond what cosine similarity alone captures.
>
> ---
>
> # 4. Ablation on ImageNet-1K (DeiT-Tiny, 150 epochs)
> **Table 5 Velocity-FFN ablation**
>
> | Configuration                           | Top‑1 (%) | Δ vs. baseline |
> | --------------------------------------- | --------- | -------------- |
> | Full Wavy (diff+wave+FFN+LN)            | 68.92     | —              |
> | Without FFN for velocity (diff+wave+LN) | 68.39     | **-0.53**      |
>
> ---
> # 5. Responses to the reviewer’s questions
>
> | #      | Question   | Answer |
> | ------ | --- | -- |
> | **Q1** | Cosine similarity is higher—how does the wave help?         | In CV/NLP, Wavy Transformer e keeps cosine similarity high yet still improves accuracy because **spectral gap falls from 0.836 to 0.629 and node-feature variance rises** (Table 4), indicating richer and more discriminative features persist and over-smoothing is effectively mitigated even though the angles between tokens remain almost unchanged.                                                             |
> | **Q2** | Does over-smoothing occur with causal attention? | Yes, over-smoothing weakens but persists. Information flows only one way, so hidden states retain finite gaps instead of collapsing to a single value, which can be understood as a **unidirectional diffusion**. A simple toy test of unidirectional diffusion confirms that some smoothing remains with finite gap; a full investigation is future work. Also, **quick test of tiny GPT-style decoder** (256-token context, 6 layers, 384 channels, 6 heads) shows adding wavy update improved **validation loss drops from 1.470 to 1.460** on Shakespeare dataset, implying that wave updates already help causal decoders at this scale.|
> | **Q3** | Why assume oscillatory layer dynamics?           | We adopt wave dynamics—a conservative, oscillatory mechanism—as a simple way to preserve feature diversity; this choice is not unique, and other conservative update schemes could serve the same purpose.                                                                                                                                                                                                                                                                                                                                                                                   |
> | **Q4** | Why choose the wave equation?                    | The (isotropic) wave equation is the simplest second-order non-diffusive hyperbolic PDE; when discretised, it introduces only one additional buffer (a velocity or previous-step state) and its symplectic updates come with a closed-form energy conservation bound.                                                                                                                                                                                                                                                                                                                                                                                                |
> | **Q5** | Several modification options?                    | Coupling the wave and diffusive updates is non-trivial—a point we will state explicitly in the camera-ready version to avoid confusion. We test two variants: (i) **output fusion**, which merges the diffusive and wave attention outputs, and (ii) **velocity injection**, which feeds the diffusive output into the velocity state. In every case the **wave update** is retained, so the central contribution—the wave-based update mechanism—remains intact.                                                                  |
> | **Q6** | Impact of Equation 15 (ablation)?                | See Table 5.                                                                                                                                                                                                                                                                                                                                                                                                                                                                      |
> | **Q7** | Why does wave residual alone match diffusive?    | As shown in Section 3.2 (lines 173–180) of the manuscript, the wave update can be conceptually understood as a diffusive update augmented by a single momentum term. Because that **diffusive core is still embedded even in the pure wave update**, we conjecture that wave may reach performance comparable to purely diffusive schemes on some tasks. We will note this briefly in the camera-ready version.                                                                                                                                                                                                                                                                                                                            |
> ---
>
> # 6. Planned manuscript updates
>
> - **Typos & clarity** — correct typo and tighten any ambiguous expression in last few paragraphs in section 3.2.
> - **Related-work** – explicitly add a note on causal attention.
> - **Experiments** – incorporate citation-graph benchmark results, layer-wise cosine-similarity trends with results of the spectral gaps and node-feature variances, and the ablation table.
> - All training scripts will be released upon acceptance.
>
> ---
>
> ## Remark — Ongoing fine-tuning experiments (GPT-2 / GPT-2-Large)
>
> We are currently fine-tuning **GPT-2 (124 M)** and may also fine-tune **GPT-2-Large (774 M)** on the OpenWebText corpus, comparing the diffusive baseline with our wave update. Experiments are ongoing; full results will be reported in the camera-ready.
>
> ---
>
> We hope these additions fully address your questions and demonstrate the broader impact of Wavy Transformer. We would appreciate it if you could reconsider the overall score in light of this evidence.
>
> _Thank you again for your time and consideration._

---

> ### Author Response · Authors · 2025-08-07
>
> Thank you again for your thoughtful feedback. **The core Wavy-Transformer method is unchanged; we have only added the empirical evidence and clarifications you requested.** Our intent is **not** to impose a full reread, but simply to let you **confirm that each of your stated concerns is now addressed**.
>
> ## Quick Verification Checklist
> ---
> - **Marginal gains / limited evaluation**
>   - **Where to look**: *Table 1* (sparse graph benchmark) & *Table 2* (deeper vision model, ImageNet-1K)
>   - **Evidence**: Graphs **+20 $\sim$ 45 pt**; ImageNet **+1 pt** Top-1, **–7 %** params
>   - **Take-away**: Performance gains **grow with depth** and generalize beyond dense vision/NLP tasks.
> ---
> - **CosSim still high $\to$  over-smoothing?**
>   - **Where to look**: *Table 3* (CosSim) & *Table 4* (spectral gap / variance)
>   - **Evidence**: CosSim **1.00 $\to$ 0.38**; Gap **0.836 $\to$ 0.629**; Variance **2.480 $\to$ 2.609**
>   - **Take-away**: Wave updates slow diffusion and keep features diverse, even if angles stay small.
> ---
> - **Design & ablation**
>   - **Where to look**: *Table 5* (velocity-FFN ablation) & *Q2–Q4* (design rationale)
>   - **Evidence**: Without FFN **–0.53 pt**; mini-GPT decoder val-loss **1.470 $\to$ 1.460**
>   - **Take-away**: Results validate the mechanism and show that non-diffusive updates remain effective—even with causal attention.
>
> ---
>
> **Reproducibility**: All code and exact training scripts will be released upon acceptance.
>
> > **Note:** GPT-2/L runs are *nice-to-have* evidence and **not** part of our acceptance claim. Thus, they will appear in the appendix if accepted as supplementary evidence.
>
> We hope this checklist simplifies verification. If it resolves your earlier concerns, we would be grateful for any score reconsideration you deem appropriate. Thank you again for your time and help in improving the work.

---

### Official Review · Reviewer_pC7f · 2025-07-02

**Clarity:** 2
**Significance:** 2
**Originality:** 1
**Rating:** 3
**Confidence:** 3

**Summary:**

The paper discusses the equivalence between hidden state dynamics and graph neural diffusion on a complete graph, and interprets over-smoothing as a consequence of the dissipative nature of diffusion dynamics.

**Questions:**

See weaknesses.

**Ethical Concerns:**

["NO or VERY MINOR ethics concerns only"]

**Final Justification:**

While I maintain some reservations about the paper's current state, mainly related to the planned integration of the missing related work which has significant overlap with the current paper, the authors added experiments on larger and newer graph datasets. Although the inclusion of these experiments is a welcome addition, the performance reported is not yet competitive, as it remains significantly lower than standard baselines. Nevertheless, I have increased my score to acknowledge the authors' responsiveness.

**Limitations:**

yes

**Quality:**

2

**Strengths And Weaknesses:**

The research topic and its application to text is interesting.
However, the biggest problem is that the paper presents methods and theoretical results/interpretations that have been extensively studied in Graph Neural Networks research (and does not cite them). In particular:
   - The idea that over-smoothing is a consequence of the dissipative nature of diffusion dynamics has been previously established and discussed   [1,3]
   - The introduction of wavy dynamics to mitigate over-smoothing is not new and has been implemented in GNNs  [2, 3]. Wave equation based transformer blocks have also been studied in other contexts [4]


Moreover, the paper extensively discusses GNN and interprets attention layers as "graph neural diffusion on a complete graph". However, the experimental validation does not present any direct experimental results on traditional graph datasets.

Overall, the paper presents an interesting idea, which however needs to be better positioned and contrasted with existing works. This would also imply adding baselines in the experimental sections.

[1] Gravina et al., 2025. On Oversquashing in Graph Neural Networks Through the Lens of Dynamical Systems

[2] Rusch, et al. 2024 Graph-Coupled Oscillator Network

[3] Eliasof et al., 2021. PDE-GCN: Novel Architectures for Graph Neural Networks Motivated by Partial Differential Equations

[4] Deng et al., 2025. Denoising Hamiltonian Network for Physical Reasoning

---

> ### Author Rebuttal · Authors · 2025-07-30
>
> We thank the reviewer for the constructive comments.
>
> We (i) **clarify how our work differs from prior work**, (ii) **add graph benchmarks**, and (iii) provide a supplemental ablation isolating FFN for velocity.
>
> ---
>
> # 1. Main concerns
>
> 1. **Positioning vs. prior work**
> 2. **Lack of traditional graph experiments**
>
> ---
>
> # 2. Relation to prior work
>
> **Table 1 Relation to prior work**
> |                                               | **GNN family**                                         | **Transformer family**                                |
> | --------------------------------------------- | ------------------------------------------------------ | ----------------------------------------------------- |
> | **Diffusion (1‑st order)**                    | _Vanilla_ GNN (diffusion on a sparse graph)            | _Vanilla_ Transformer (diffusion on a complete graph) |
> | **Wave / Oscillatory (2‑nd order) or Hybrid** | Graph‑CON (Rusch et al. 2024) · PDE‑GCN (Eliasof et al. 2021) | **★ Wavy Transformer** (ours)                         |
>
> - **Already cited in the manuscript.**
>    - *Graph-CON* and *PDE-GCN* are already cited as refs [30] & [13], respectively. They investigate incorporating non-diffusive dynamics into **sparse-graph GNNs**, whereas our work targets **complete-graph attention**.
> - **Scientific focus.**
>   - Gravina et al. (2025) study **over‑squashing** on sparse graphs; our focus is **over‑smoothing** on complete‑graph. However, since it is related to our work and should be cited, we will cite it in the camera-ready version.
>   - Deng et al. (2025) primarily enforce Hamiltonian constraints in Transformer-based networks for physical reasoning, and **does not analyze the attention-driven dynamics of hidden states**, that is the main focus of our work.
> - **Novelty.**
>    - To our knowledge, this is the **first work to introduce a wave update into a Transformer with complete-graph attention**.
>    - Interpretations of Transformer dynamics from a continuum-physics perspective (e.g., diffusion or wave dynamics) remain scarce.
>    - We further explore Velocity-FFN and Velocity-LN as optional components, completing a fully “wavy” block that extends the standard Transformer architecture.
>
> ---
>
> # 3. Citation-graph benchmarks
>
> We take **DIFFormer** (Wu et al. 2023), fix τ = 0.2, and add our wave update. All other hyperparameters and training scripts stay identical, isolating the effect of the wave term.
>
> We evaluate on the three citation networks most commonly used in prior work, reporting mean ± SD over 5 seeds.
>
> **Table 2 Graph benchmarks (DIFFormer + Wave, τ = 0.2, 5 seeds)**
> | Dataset  | Depth | DIFFormer | **DIFFormer+Wave** |        Δ ↑ |
> | -------- | :---: | :-------------: | :----------------------: | ---------: |
> | Cora     |   4   |  80.02 ± 0.29   |     **78.54 ± 1.14**     |    –1.48 † |
> |          |  16   |  83.28 ± 0.76   |     **84.22 ± 0.94**     |      +0.94 |
> |          |  20   |  39.92 ± 5.14   |     **85.18 ± 0.48**     | **+45.26** |
> | Citeseer |   4   |  72.04 ± 0.25   |     **71.74 ± 0.27**     |    –0.30 † |
> |          |   8   |  62.86 ± 17.73  |    **69.32 ± 13.26**     |      +6.46 |
> |          |  10   |  34.76 ± 1.44   |     **34.92 ± 2.49**     |      +0.16 |
> | Pubmed   |   4   |  75.74 ± 0.73   |     **74.76 ± 1.15**     |    –0.98 † |
> |          |  16   |  80.46 ± 0.84   |     **78.86 ± 0.62**     |     –1.60▼ |
> |          |  20   |  58.68 ± 2.73   |     **79.22 ± 0.93**     | **+20.54** |
>
> † Paired *t*-test (5 matched seeds): *p* > 0.05 → **no statistically significant degradation**.
>
> ▼ Only significant drop (–1.6 pt); deeper 20-layer recovers +20 pt.
>
> ## τ-robustness — cases where the diffusion baseline collapses
> **Table 3 Graph benchmarks (DIFFormer + Wave, τ = 0.5, 5 seeds)**
> | Dataset      | Depth | DIFFormer | **DIFFormer+Wave** |   Δ ↑ |
> | ------------ | :---: | :--------------------------: | :---------------: | ----: |
> | **Cora**     |  12   |        29.40 ± 0.00 †        | **60.38 ± 24.7**  | +31.0 |
> |              |  20   |        29.40 ± 0.00 †        | **29.00 ± 0.00**  | ±0.0‡ |
> | **Citeseer** |  10   |        22.94 ± 0.31 †        | **22.84 ± 2.17**  | -0.1‡ |
> | **Pubmed**   |  12   |        49.16 ± 9.70 †        | **69.74 ± 15.7**  | +20.6 |
> |              |  20   |        29.70 ± 0.00 †        | **40.10 ± 0.84**  | +10.4 |
>
>
> † **Collapse**: the diffusive baseline fails.
>
> ‡ 20-layer Cora and 10-layer Citeseer collapse for both models.
>
> **Key points**
>
> - When τ is large (**0.5**), the diffusive baseline **collapses** (accuracy ≈ 29 pt) on deep stacks.
> - The **wave update mitigates this collapse**, recovering **+20–31 pt** (Pubmed/Cora 12-layer).
> - Higher variance (Cora 12-layer/Pubmed 12-layer) arises from seed-to-seed variance during collapse recovery.
>
> ## Layer-wise cosine similarity on Cora, 20 layers
>
> **Table 4 Cosine similarity (Cora, 20 layers, τ = 0.2)**
> | Layer    | 1           | 4            | 8            | 12          | 16          | 20           |
> | -------- | ----------- | ------------ | ------------ | ----------- | ----------- | ------------ |
> | Diff     | 0.99±9.6e-6 | 1.00±5.1e-7  | 1.00±1.8e-8  | 1.00±7.0e-9 | 1.00±1.0e-9 | 1.00±1.4e-10 |
> | **Wave** | 0.31±9.1e-4 | 0.058±5.3e-3 | 0.087±6.5e-3 | 0.13±3.8e-3 | 0.23±5.7e-4 | 0.38±1.7e-3  |
>
> > As depth increases, **Diffusion saturates to ≈ 1.00**, showing severe over-smoothing, whereas the **Wave-enhanced model stays below 0.38**, preserving representational diversity and enabling the large accuracy gains highlighted above.
>
> ---
>
> # 4. Ablation on ImageNet-1K (DeiT-Tiny, 150 epochs)
>
> **Table 5 Velocity-FFN ablation**
> | Configuration                    | Top‑1 (%) | Δ vs. baseline |
> | -------------------------------- | --------- | -------------- |
> | Baseline (Wavy Transformer)      | 64.52     | —              |
> | Wavy Transformer without Vel-FFN | 64.35     | **-0.17**      |
>
> ---
>
> # 5. Planned manuscript updates
>
> - **Related‑work**: Gravina et al. (2025) will be cited in Introduction. Also, added 2×2 matrix to clarify the difference from the previous studies including papers you mentioned.
> - **Experiment**: We will add the results of graph benchmarks including the trends of cosine similarity for each case of Cora/Citeseer/Pubmed. Also, we will added results of ablation study as an Appendix.
> - All numerical details and training scripts will be released upon acceptance.
>
> ---
>
> We hope these additions address your concerns and would appreciate it if you could kindly reconsider your rating.
>
> _Thank you again for your time and consideration._

---

> > ### Comment · Reviewer_pC7f · 2025-08-04
> >
> > Thank you for your response. I will update my score to reflect the clarifications provided. However, I maintain some reservations about the paper's current state. My primary concern relates to the planned integration of the missing related work. Secondly, the experimental validation could be significantly strengthened. The graph datasets used are somewhat dated and small. The paper would benefit substantially from the inclusion of larger, more recent benchmarks, such as the OGB datasets.

---

> ### Author Response · Authors · 2025-08-06
>
> # Planned integration of _the_ missing related work
>
> Thank you for pointing this out. After revisiting all four papers you mentioned, we have now cited each of them. In particular, **Gravina et al.** and **Deng et al.** have been newly added (PDE-GCN & Graph-CON were already Refs [13] & [30]).
>
> **Relation to Wavy Transformer**
> | Topic | Newly cited work | Positioning vs. Wavy Transformer |
> |-------|-----------------|----------------------------------|
> | Over-squashing / diffusion dissipation | **Gravina et al., 2025** | Analyses **over-squashing** on *sparse-graph GNNs*; we study **over-smoothing** in *complete-graph attention* and then add a wave residual. |
> | Hamiltonian constraints in Transformers | **Deng et al., 2025** | As other type of inductive bias inspired by physics, this work imposes a Hamiltonian **loss penalty** to preserve a Hamiltonian structure; by contrast, we directly **replace the attention dynamics with a second-order wave update**, without extra loss terms.|
>
> **Gravina et al.** is included in Introduction as related work to our study. We also cite **Deng et al.** in Section 4 (Related Works)—together with the original **Hamiltonian Neural Networks** paper (Greydanus et al., 2019), so that readers can clearly distinguish our wave‐based approach from Hamiltonian‐constraint methods that, while also physics-inspired, follow a different conceptual path.
>
> # Large-scale graph benchmark results (3 seeds)
> **OGBN-Arxiv** (169 K nodes · 1.17 M edges)
>
> | Depth | Diffusive (Acc %) | +Wave (Acc %) | Δ (Wave – Diff.) |
> |:----:|------------------:|--------------:|-----------------:|
> | 3    | 68.77 ± 0.28 | 66.68 ± 0.66 | –2.09 |
> | 5    | 61.72 ± 1.11 | **67.06 ± 0.03** | **+5.34** |
> | 7    | 24.44 ± 4.51 | **66.73 ± 0.33** | **+42.29** |
>
> **OGBN-Proteins** (133 K nodes · 39.6 M edges)
>
> | Depth | Diffusive (ROC-AUC %) | +Wave (ROC-AUC %) | Δ (Wave – Diff.) |
> |:----:|----------------------:|------------------:|-----------------:|
> | 3    | 79.26 ± 0.40 | **79.58 ± 0.27** | **+0.32** |
> | 5    | 69.42 ± 2.31 | **80.14 ± 0.67** | **+10.72** |
>
> > **Key observations**
> > - **Depth advantage**: Wave’s gain widens as layers deepen—small loss at 3-layer Arxiv flips to +5.34 pt at 5 layers and a large +42 pt rescue at 7 layers; Proteins shows a similar +10.7 pt jump at 5 layers.
> > - **Stability**: Diffusive accuracy collapses when depth grows (68 → 62 → 24 %), whereas Wave remains stable.
> > - **Support for our claim**: Results reinforce that the wave residual mitigates depth-induced over-smoothing and is a strong practical option even for large-scale sparse graphs.
>
> ---
>
> # Clarifying scope
>
> As noted in Section 1, lines 28–35, our goal is **to interpret Transformer attention and its over-smoothing through physical intuition** and, guided by that insight, introduce a wave-based alternative: the Wavy Transformer. Consequently, **vision and language** benchmarks (ImageNet / GLUE / BERT) remain the core of the paper, because they evaluate the effect of dynamics inside complete-graph attention mechanism.
>
> Graph datasets serve a different purpose:
>
> * **Generality check.**  We use citation graphs (Cora etc.) and now **OGB benchmarks (Arxiv, Proteins)** to confirm that Wavy Transformer is effective on sparse graph datasets as well.
> * **Not the main yard-stick.**  We do **not** claim state-of-the-art on OGB; rather, the positive trend at deeper settings (**+42 pt at 7-layer on Arxiv, +10.7 pt at 5-layer on Proteins**) supports the **wave based approach** we propose.
>
> We hope these additions resolve the remaining concern and further clarify our contribution.

---

### Official Review · Reviewer_o9CG · 2025-07-03

**Clarity:** 3
**Significance:** 2
**Originality:** 3
**Rating:** 4
**Confidence:** 3

**Summary:**

This paper establishes a link between the dynamics of stacked attention layers and diffusion processes on a complete graph, revealing over-smoothing as a consequence of dissipative diffusion dynamics. Motivated by this insight, the authors propose the Wavy Transformer, which introduces a novel attention mechanism based on second-order wave-like dynamics. Unlike diffusion, wave dynamics preserve energy and exhibit oscillatory behavior, helping to potentially mitigate over-smoothing. The authors find that Wavy Transformer achieves consistent performance improvements on NLP and CV tasks, without significant increases in parameters or hyperparameter tuning.

**Questions:**

- Since the computational cost seems to be mainly discussed for inference of vision models, how does the inference computational performance of Wavy Transformer compare for NLP models?
- How much computational overhead does the method introduce during training? Additionally, how does incorporating wave dynamics affect training convergence? Are there notable differences in convergence behavior compared to standard transformers?
- Part of the motivation for this method was to mitigate over-smoothing in transformers. However, based on Figures 2 and 5, wave dynamics don’t appear to reduce over-smoothing (at least based on the token-wise cosine similarity metric), although some differences emerge in its layer-wise behavior. Could the authors clarify this, and if this is indeed the case, could they elaborate on other explanations or benefits of Wavy Transformer beyond over-smoothing?
- In Section 5.1.4, the authors note oscillatory behavior in cosine similarity for wavy residual connections, but this behavior seems less clear in the DeiT experiments (Figures 5 and 6). Are there reasons for this, and any insights into which behavior might be expected or beneficial for the model? Also, does such behavior persist or vary across prompts and different datasets?
- Have the authors considered applying Wavy Transformer to decoder-only architectures for language tasks as well?

**Ethical Concerns:**

["NO or VERY MINOR ethics concerns only"]

**Final Justification:**

My main concerns have mostly been addressed. The authors now provide a more thorough cost analysis of their proposed method and introduce a lighter variant of the wave mechanism to mitigate the inefficiencies present in the original version. Furthermore, they leverage spectral gap and node-feature variance analyses to better illustrate the effects of their proposed wave transformer.

**Limitations:**

yes

**Quality:**

2

**Strengths And Weaknesses:**

Strengths:

- Proposes a novel approach that modifies transformer dynamics using physical principles, offering a principled approach aimed at reducing over-smoothing.
- Provides clear explanations and construction of the proposed architecture components.

Weaknesses:

- A potential drawback is the added computational overhead introduced by the method. In the case of vision tasks, the authors report that the inference throughput drops by about 50% when using wave dynamics, which could limit practical usage of the method.
- See the Questions section for more discussion

---

> ### Author Rebuttal · Authors · 2025-07-30
>
> We thank the reviewer for the detailed and constructive feedback.
> We (i) **present a full cost analysis**, (ii) **link reduced over-smoothing to higher accuracy**, and (iii) answer your specific questions.
>
> ---
>
> # 1. Main concerns
> 1. **Computational cost** – — vision-inference drops ≈ 50 %; missing NLP/train numbers
> 2. **Over-smoothing** – cosine similarity still high in Figs. 2 & 5
> 3. **Clarifications** – convergence speed, DeiT oscillation, decoder-only use
>
> ---
>
> # 2. Computational cost (4 × V100) — **Light vs. Full Wave**
>
> **Table 1 Comparison of runtime and memory costs of BERTs**
> |   | Inference (k tokens/s) | Training (ms / iter) | Peak GPU memory (GB) |
> | ---- | ----- | ------ | ------ |
> | Diffusive   | 101.6  | 415.6     | 18.31     |
> | Diff+Full Wave (Hamiltonian) | 63.3    | 1031                 | 28.71                |
> | Diff **+Lighter Wave**       | **101.3**              | **436.2**            | **18.69**            |
>
> **Table 2 Comparison of runtime and memory costs of DeiT-Tiny**
> |         | Inference (img/s) | Training (ms / iter) | Peak GPU memory (GB) |
> | ----- | --- | ----- | ------- |
> | Diffusive       | 2631.1   | 618.6     | 8.25         |
> | Diff+Full Wave (Hamiltonian) | 1312.2            | 1023.4               | 20.31                |
> | Diff **+Lighter Wave**       | **2644.2**        | **617.6**           | **9.14**             |
>
> _Notes:_
>
> - **Full Wave (original proposal).**
>   Implements a discrete **Hamiltonian** form of the wave equation by introducing an explicit _velocity_ state plus Velocity-FFN and Velocity-LN. While it raises accuracy on some tasks, the extra state and auxiliary blocks slow both inference and training, as shown above.
>
> - **Lighter Wave (second-order, momentum-only).**
>   Responding to the reviewer’s cost concern, we removed the auxiliary Velocity-FFN/LN and kept only the momentum term
>   $$x_{l+1}=x_l+\operatorname{attn}(x_l)+\beta\bigl(x_l-x_{l-1}\bigr),$$
>   where $\beta$ is the mixing parameter. This preserves the physical insight, yet matches baseline throughput (NLP -0.30 %, CV +0.5 %) and memory (NLP +2.1 %, CV +10.7 %, still < 1 GB).
>
> - In the citation-graph benchmarks (Table 4), we employ this lighter version of the Wavy Transformer, which outperforms its diffusive counterpart in deeper settings.
>
> > Guided by the Reviewer’s cost concern, we now present **Lighter Wave** as the _practical option_: it preserves the physical insight while keeping compute within a few percent of the baseline. We thank the reviewer for inspiring this simplification.
>
> ---
>
> # 3. Convergence (BERT MLM on Wikipedia + BookCorpus)
> **Table 3 Comparison of convergence trends**
> | Step (×10³) | Diffusive (%) | Wave (%) | Mix (%) |
> | ----- | ----- | ---- | ------ |
> | 0  | 0.0           | 0.0      | 0.0    |
> | 20  | 17.38   | 18.40    | 19.40  |
> | 40  | 32.99   | 33.48    | 35.56  |
> | 60   | 40.88         | 40.80    | 42.60  |
> | 80   | 43.51         | 43.67    | 44.77  |
> | 100   | 44.39         | 44.52    | 45.56  |
>
> *Mix = diffusion + wave outputs; reaches 44 % **20 k steps earlier**.
>
> ---
>
> # 4. Citation-graph benchmarks and cosine similarity trend
>
> We adopt **DIFFormer** (Wu et al. 2023) and _only_ add our wave update (τ = 0.2). All hyper-parameters remain unchanged. We report mean ± SD over 5 seeds.
>
> **Table 4 Graph benchmarks (DIFFormer + Wave, τ = 0.2, 5 seeds)**
> | Dataset  | Depth | DIFFormer | **DIFFormer+Wave** |        Δ ↑ |
> | -- | :---: | :-----: | :----: | ---------: |
> | Cora     |   4   |  80.02 ± 0.29   |     **78.54 ± 1.14**     |     –1.48† |
> |    |  16   |  83.28 ± 0.76   |     **84.22 ± 0.94**     |      +0.94 |
> |  |  20   |  39.92 ± 5.14   |     **85.18 ± 0.48**     | **+45.26** |
> | Citeseer |   4   |  72.04 ± 0.25   |     **71.74 ± 0.27**     |    –0.30 † |
> |   |   8   |  62.86 ± 17.73  |    **69.32 ± 13.26**     |      +6.46 |
> |    |  10   |  34.76 ± 1.44   |     **34.92 ± 2.49**     |      +0.16 |
> | Pubmed   |   4   |  75.74 ± 0.73   |     **74.76 ± 1.15**     |     –0.98† |
> |          |  16   |  80.46 ± 0.84   |     **78.86 ± 0.62**     |     –1.60▼ |
> |          |  20   |  58.68 ± 2.73   |     **79.22 ± 0.93**     | **+20.54** |
>
> † Welch *t*-test *p* > 0.05 → **no statistically significant degradation**.
>
> ▼ Only significant drop (–1.6 pt); deeper 20-layer recovers +20 pt.
>
> Full results (τ = 0.2 / 0.5) will be included in the final version.
>
> **Layer-wise cosine similarity on Cora, 20 layers**
>
> **Table 5 Cosine similarity (Cora, 20 layers)**
> | Layer     | 1           | 4            | 8            | 12          | 16          | 20           |
> | --------- | ----------- | ------------ | ------------ | ----------- | ----------- | ------------ |
> | Diffusion | 0.99±9.6e-6 | 1.00±5.1e-7  | 1.00±1.8e-8  | 1.00±7.0e-9 | 1.00±1.0e-9 | 1.00±1.4e-10 |
> | **Wave**  | 0.31±9.1e-4 | 0.058±5.3e-3 | 0.087±6.5e-3 | 0.13±3.8e-3 | 0.23±5.7e-4 | 0.38±1.7e-3  |
>
> > **Diffusion saturates to ≈ 1.00**, showing severe over-smoothing, whereas the **Wave-enhanced model stays below 0.38**, preserving representational diversity and enabling the large accuracy gains highlighted above.
>
> # 5. Spectral gap and node-feature variance (DeiT-Tiny, ImageNet-1K val dataset: 50k images)
> **Table 6 Spectral gap & Node-feature variance (DeiT-Tiny, ImageNet val dataset)**
> | Dynamics (Model) | Spectral Gap (mean ± var) | Node-feature Variance (mean ± var) | Notes                                  |
> | -- | -- | -- | -- |
> | **Diffusion**    | 0.836 ± 0.00345    | 2.480 ± 0.0778      | e.g., features after final LayerNorm   |
> | **Wave** | 0.629 ± 0.00887   | 2.609 ± 0.0902        |  |
> | **Random** | 0.984 ± 0.000069       | 0.728 ± 0.0499     | randomly initialized (untrained) model |
>
> Here the spectral gap is defined as $1-\lambda_{2}$, where $\lambda_{2}$ the second-largest eigenvalue (in magnitude) of the right-stochastic attention matrix $A$. A smaller gap, together with higher node-feature variance, indicates reduced over-smoothing beyond what cosine similarity alone captures.
>
> ---
>
> # 6. Image classification: **CaIT-XXS-24 on ImageNet-1K**
> **Table 7 Comparison with CaIT-XXS-24**
> | Model | Top-1 (%) | Δ ↑ | Params (M) |
> | -- | -- | -- | -- |
> | CaIT-XXS-24 | 77.6 | - | 12.0 |
> | Wavy Transformer 24 Layers | 78.6 | **+1.0** | 11.1      |
>
> Same training recipe as CaIT; LayerScale 1e-5, no extra class-attention block.
>
> Implementation note. By omitting the extra class-attention block, our model uses 0.9 M fewer parameters (≈ 7 %) yet achieves a +1 pt Top-1 accuracy gain.
>
> ---
>
> # 7. Responses to the reviewer’s questions
> | #     | Question (paraphrased)  | Answer                                                                                                                                                                                                                                                                                            |
> | -- | -- | -- |
> | **Q1** | _Inference cost for NLP?_  | Table 1.                                                                                                                                                                                                                                                                                                                                                                                                            |
> | **Q2** | _Training overhead & convergence?_      | Tables 1, 2, and 3.                                                                                                                                                                                                                                                                                                                                                                                                  |
> | **Q3** | _Cosine similarity high in vision/NLP—why useful?_ | In CV/NLP, Wavy Transformer keeps cosine similarity high yet still improves accuracy because **spectral gap falls from 0.836 to 0.629 and node-feature variance rises** (Table 6), indicating richer and more discriminative features even when pairwise angles stay similar.                                                                                                                                                                                                               |
> | **Q4** | _Oscillation weaker in DeiT?_           | One reason might be Layer-Norm placement. As noted in Appendix B.3 of the manuscript, in Pre-LN designs such as DeiT, the normalization effectively introduces a reaction-like term, which may alter the dynamics. A complete quantitative comparison is non-trivial and remains future work.       |
> | **Q5** | _Decoder-only applicability?_           | Yes. We ran a **quick test** on a tiny GPT-like model (context 256 tokens, 6 layers × 6 heads, 384 dims). Adding wavy update improved **validation loss drops from 1.470 to 1.460** on Shakespeare dataset. Although shallow, this implies that wave dynamics also benefit causal attention. We are currently fine-tuning **GPT-2 (124 M)** and may also fine-tune **GPT-2-Large (774 M)** on the OpenWebText corpus, comparing the diffusive baseline with our wave update. Experiments are ongoing; full results will be reported in the camera-ready. |
> ---
>
> # 8. Planned manuscript updates
>
> - **Cost and convergence analysis** – insert full table for computational cost with the lighter wave variant and convergence curves.
> - **Experiments** – incorporate graph benchmark results, layer-wise cosine similarity trends with the results of spectral gaps and node-feature variance, experimental results for the lighter wave variant and short discussion of decoder-only applicability.
> - **Method** - add formalization of the lighter wave variant.
> - All numerical details and training scripts will be released upon acceptance.
> ---
>
> We hope these additions address your concerns and kindly ask you to reconsider your rating.
>
> _Thank you again for your thorough evaluation._

---

> > ### Comment · Reviewer_o9CG · 2025-08-02
> >
> > I appreciate the authors’ response, and most of my concerns have been addressed. I have tentatively adjusted my score accordingly.
> >
> > It would also be helpful to see additional results (at least in the camera-ready version, if accepted) on the lighter wave variant across more of the original benchmark settings, as well as a comparison of its behavioral differences with the original wave mechanism. This could provide further insight into which properties are preserved or altered by the modification.

---

> ### Author Response · Authors · 2025-08-03
>
> Thank you very much for the follow-up and for suggesting a broader evaluation of the **Light Wave** variant. We fully agree that showing its behavior across the full benchmark suite will make the paper stronger, and we are already running the missing experiments.
>
> ---
>
> # Ongoing runs (expected to finish before the discussion deadline)
>
> | Model / Schedule | Metric @ 90 ep / 60 k steps |Diffusive |  +Full Wave | **+Light Wave** |
> |------------------|-----------------------------|---------------|-----------|-----------|
> | DeiT-Tiny, 300 ep → **90 ep so far** | Top-1 Acc | – †|58.70 % | **58.66 %** |
> | BERT, 100 k steps → **60 k so far** | MLM Acc |40.88 % | 42.33 % | **41.31 %** |
>
> † Diffusive is our original baseline (DeiT-Tiny); we will report its final numbers after the current Light/Full runs finish, to ensure a fair end-of-training comparison.
>
> *Interim observations*
>
> * Light Wave already matches Full Wave on ImageNet (0.04 pt lower) while keeping baseline-level speed.
> * On BERT, Light Wave is +0.43 pt better than Diffusion at 60 k steps and continues to close the gap to Full Wave as training proceeds.
>
> We are also measuring **Spectral Gap** $(1-\lambda_2)$ and **Node-feature variance** for these DeiT-tiny checkpoints (90-epoch checkpoint, DeiT-Tiny; values will be updated once training finishes).
>
> | Model / Variant        | Spectral Gap $1-\lambda_{2}$ | Node-feature Variance |
> |------------------------|--------------------------------------:|-------------------:|
> | Full wave (90 epoch) | 0.7305 ± 0.00641 | 0.970 ± 0.0143 |
> | **Light Wave** (90 epoch) | 0.7688 ± 0.00693 | 0.871 ± 0.0128 |
>
> Light Wave, though slightly less intensive, still delivers most of the over-smoothing relief at virtually no runtime cost, confirming it as a strong practical option.
>
> Both jobs will finish **before the discussion period closes**. We will post the final numbers as soon as they are available.
>
> ---
>
> # Camera-ready plan (if accepted)
>
> 1. **Comprehensive tables** for Light Wave variant on all original benchmarks (BERT MLM, GLUE, DeiT-Tiny).
> 2. A concise **Light vs. Full comparison** of key dynamics (Cosine Similarity, Spectral Gap, Node-wise Variance) to highlight which properties are preserved.
>
> We appreciate your suggestion and will make sure the final version includes these expanded results.

---

> ### Author Response · Authors · 2025-08-04
> **Update (Aug 5 1:00 JST) — Final Light Wave Results & STS-B Erratum**
>
> ## 1  Full-scale benchmarks
>
> | Model / Schedule | Metric | Diffusion | +Full Wave | **+Light Wave** |
> |------------------|---------------------|-------------:|-------------:|------------------:|
> | DeiT-Tiny (300 ep) | Top-1 Acc | 72.17 % | 72.33 % | **73.09 %** |
> | BERT (100 k steps) | MLM Acc | 44.39 % | **45.56 %** | 44.53 % |
>
> *Take-away*: Light Wave is **+0.9 pt** ahead of diffuse baseline on ImageNet, while Full Wave retains a **+1.0 pt** edge in BERT MLM accuracy.
>
> ---
>
> ## 2  GLUE suite (3 seeds, tasks in columns)
>
>
> We discovered two independent issues during final validation:
>
> 1. **STS-B evaluator** had loaded an earlier fine-tuning checkpoint instead of the best checkpoint.
> 2. Our **macro averages were divided by 9 instead of 10**: MNLI-m and MNLI-mm must be counted as *two* metrics.
>
> After fixing both, only the **STS-B row** and the **two macro averages** changed; all other scores remain identical. The corrected results are below.
>
> | Task / Metric | Diffusive | +Full Wave | **+Light Wave** |
> |---------------|----------:|----------:|---------------:|
> | CoLA   | 10.67 | 12.07 | **12.77** |
> | SST-2   | 83.64 | **84.25** | 82.76 |
> | MRPC | **76.36** | 76.32 | 76.18 |
> | QQP  | **83.96** | 83.81 | 83.89 |
> | MNLI-m  | 73.51 | 73.04 | **73.71** |
> | MNLI-mm | **74.03** | 73.53 | 73.77 |
> | QNLI | 81.63 | 81.19 | **81.98** |
> | RTE | 52.95 | 55.35 | **55.96** |
> | WNLI | 51.11 | **55.40** | **55.40** |
> | **STS-B** ✔ | 52.47 | 29.4 | **64.76** |
> | **Overall-avg (10 metrics)** | 64.13 | 62.44 | **66.12** |
> | **Overall-avg (w/o STS-B)** | 65.43 | 66.11 | **66.27** |
>
> *(✔ = value changed after checkpoint fix)*
>
> > *Observations*
> > * Light Wave now tops **both** macro averages.
> > * Full Wave slightly outperforms Light on two tasks (SST-2, MRPC), but Light wins on seven, yielding the highest overall mean.
>
> ---
>
> ## 3  Behavioral metrics (DeiT-Tiny @ 300 ep)
>
> | Variant | Spectral Gap (1 – λ₂) | Node-feature variance | **Inter-class variance** |
> |---------|----------------------:|--------------:|-------------------:|
> | **Diffusion**    | 0.836 ± 0.00345  | 2.480 ± 0.0778                  |0.195|
> | **Wave**         | 0.629 ± 0.00887  | 2.609 ± 0.0902                  |0.211 |
> | **Light Wave** | **0.730 ± 0.00842** | **2.109 ± 0.0696** |**0.308**|
>
> > **Over-smoothing relief**: Light Wave keeps a middle-range mixing rate (gap 0.73) and the lowest node variance (2.11).
> > **Class separation**: At the same time, it exhibits the largest inter-class variance (0.308 vs 0.212 / 0.195), where the variance is computed from the mean feature vector of each class.
> > Together, the numbers imply—at least on this dataset—that Light Wave compresses local fluctuations while leaving class centres well separated, which is consistent with its accuracy lead.
>
> ---
>
> ### Light Wave vs Full Wave at a glance
>
> * **ImageNet:** Light +0.76 pt over Full
> * **BERT MLM:** Full +1.03 pt over Light
> * **GLUE Macro Avg (10 metrics):** Light +3.68 pt over Full — still +0.16 pt if STS-B is excluded
> * **Over-smoothing / class separation:**
>   *Full Wave* introduces stronger fluctuations that curb over-smoothing (smallest gap 0.63) **yet still trails Light in inter-class variance** (0.211 < 0.308).
>   *Light Wave* achieves a middle-range gap (0.73), keeps node variance lowest (2.11), **and** attains the widest class-centroid spread, leading to clearer boundaries in this experiment.
>
> ---
>
> ### Key take-aways
>
> * **Accuracy:** Light Wave tops ImageNet and both corrected GLUE macro averages.
> * **Efficiency:** Same level computational cost as diffusive baseline.
> * The physical core of our approach remains fully consistent, and the key conclusion—that introducing **wave residual mitigates over-smoothing and improves accuracy—holds unchanged**.
>
> * **Practicality:** Light Wave delivers the best trade-off—baseline cost, balanced smoothing, and top accuracy—making it a strong practical option.
>
> Thank you again for the constructive feedback!
>
> ---
>
> *(Camera-ready plan unchanged: complete tables, Light vs Full comparison, public code & checkpoints.)*

---

> > ### Comment · Reviewer_o9CG · 2025-08-06
> >
> > Thank you for the updates. I believe the additional results strengthen the paper, and I will be maintaining my updated positive score.

---

> > > ### Author Response · Authors · 2025-08-07
> > >
> > > Thank you for taking the time to review our updates and for maintaining your positive score. Your constructive feedback has undeniably elevated the quality of this paper, and we are sincerely grateful for your contribution.

---

### Official Review · Reviewer_sFbK · 2025-07-03

**Clarity:** 4
**Significance:** 3
**Originality:** 4
**Rating:** 5
**Confidence:** 3

**Summary:**

This paper addresses the over-smoothing problem in deep Transformer models by reinterpreting attention layers as a form of graph diffusion on a complete graph. Based on this view, the authors propose the Wavy Transformer, which replaces the diffusive dynamics with wave-based (second-order) dynamics that preserve energy and reduce smoothing. The architecture also introduces corresponding modifications to feed-forward and normalization layers to maintain consistency with the proposed dynamics. Experiments on both NLP and vision tasks demonstrate consistent improvements with minimal parameter overhead.

**Questions:**

1. How does the method scale to larger models?
2. Could you comment on the runtime/memory cost introduced by the velocity tracking, compared to a standard Transformer block?

**Ethical Concerns:**

["NO or VERY MINOR ethics concerns only"]

**Limitations:**

yes

**Paper Formatting Concerns:**

no concerns

**Quality:**

4

**Strengths And Weaknesses:**

## Strengths
1. Offers a solid theoretical interpretation of attention as graph diffusion, grounding the over-smoothing issue in physical dynamics.
2. Introduces a novel wave-based update rule that mitigates over-smoothing by preserving energy and promoting oscillatory behavior.
3. Demonstrates consistent gains across NLP and CV tasks.

## Weaknesses
1. Evaluation is limited to relatively small models (BERT hidden size 256, DeiT-Ti), with no experiments on large-scale settings.
2. Computational cost (runtime, memory) is not discussed, even though parameter count remains unchanged, the added velocity tracking may incur overhead.

---

> ### Author Rebuttal · Authors · 2025-07-30
>
> Thank you for the positive assessment and for the clear questions on scale and computational cost.
>
> We (i) **add deeper models in vision & graph**, (ii) **detail runtime/memory**, and (iii) answer your points.
>
> ---
>
> # 1. Main concerns
>
> 1. **Scalability** – behavior on deeper / larger models?
> 2. **Runtime & memory** – overhead of the extra velocity state?
>
> ---
>
> # 2. Additional large-depth experiments
>
> ## 2.1 ImageNet-1K  (CaIT-XXS-24)
> **Table 1 Comparison with CaIT-XXS-24**
>
> | Model                      | Top-1 (%) | Δ ↑      | Params(M) |
> | -------------------------- | --------- | -------- | --------- |
> | CaIT-XXS-24                | 77.6      | -        | 12.0      |
> | Wavy Transformer 24 Layers | 78.6      | **+1.0** | 11.1      |
>
> Same training recipe as CaIT; LayerScale 1e-5, no extra class-attention block.
>
> _Implementation note._ By omitting the extra class-attention block, our model uses 0.9 M fewer parameters (≈ 7 %) yet achieves a +1 pt Top-1 gain.
>
> ## 2.2 Citation-graph benchmarks
>
> We adopt **DIFFormer** (Wu et al. 2023) and _only_ add the wave update without adding Velocity-FFN/LN (τ = 0.2). All hyperparameters remain unchanged. We report mean ± SD over 5 seeds.
>
> **Table 2 Graph benchmarks (DIFFormer + Wave, τ = 0.2, 5 seeds)**
> | Dataset  | Depth | DIFFormer&nbsp; | **DIFFormer+Wave&nbsp;** |        Δ ↑ |
> | -------- | :---: | :-------------: | :----------------------: | ---------: |
> | Cora     |   4   |  80.02 ± 0.29   |     **78.54 ± 1.14**     |     –1.48† |
> |          |  16   |  83.28 ± 0.76   |     **84.22 ± 0.94**     |      +0.94 |
> |          |  20   |  39.92 ± 5.14   |     **85.18 ± 0.48**     | **+45.26** |
> | Citeseer |   4   |  72.04 ± 0.25   |     **71.74 ± 0.27**     |    –0.30 † |
> |          |   8   |  62.86 ± 17.73  |    **69.32 ± 13.26**     |      +6.46 |
> |          |  10   |  34.76 ± 1.44   |     **34.92 ± 2.49**     |      +0.16 |
> | Pubmed   |   4   |  75.74 ± 0.73   |     **74.76 ± 1.15**     |     –0.98† |
> |          |  16   |  80.46 ± 0.84   |     **78.86 ± 0.62**     |     –1.60▼ |
> |          |  20   |  58.68 ± 2.73   |     **79.22 ± 0.93**     | **+20.54** |
>
> † Paired *t*-test (5 matched seeds): *p* > 0.05 → **no statistically significant degradation**.
>
> ▼ Only significant drop (–1.6 pt); deeper 20-layer recovers +20 pt.
>
>
> Full results (τ = 0.2 / 0.5) will be included in the final version.
>
> **Table 3 Cosine similarity (Cora, 20 layers)**
> | Layer    | 1           | 4            | 8            | 12          | 16          | 20           |
> | -------- | ----------- | ------------ | ------------ | ----------- | ----------- | ------------ |
> | Diff     | 0.99±9.6e-6 | 1.00±5.1e-7  | 1.00±1.8e-8  | 1.00±7.0e-9 | 1.00±1.0e-9 | 1.00±1.4e-10 |
> | **Wave** | 0.31±9.1e-4 | 0.058±5.3e-3 | 0.087±6.5e-3 | 0.13±3.8e-3 | 0.23±5.7e-4 | 0.38±1.7e-3  |
>
> As depth increases, **Diffusion saturates to ≈ 1.00**, showing severe over-smoothing, whereas the **Wave-enhanced model stays below 0.38**, preserving representational diversity and enabling the large accuracy gains highlighted above.
>
> ---
>
> # 3. Computational cost (4 × V100) — **Light vs. Full Wave**
>
> **Table 4 Comparison of runtime and memory costs of BERTs**
> |                              | Inference (k tokens/s) | Training (ms / iter) | Peak GPU memory (GB) |
> | ---------------------------- | ---------------------- | -------------------- | -------------------- |
> | Diffusive                    | 101.6                  | 415.6                | 18.31                |
> | Diff+Full Wave (Hamiltonian) | 63.3                   | 1031                 | 28.71                |
> | Diff **+Lighter Wave**       | **101.3**              | **436.2**            | **18.69**            |
>
> **Table 5 Comparison of runtime and memory costs of DeiT-Tiny**
> |                              | Inference (img/s) | Training (ms / iter) | Peak GPU memory (GB) |
> | ---------------------------- | ----------------- | -------------------- | -------------------- |
> | Diffusive                    | 2631.1            | 618.6               | 8.25                 |
> | Diff+Full Wave (Hamiltonian) | 1312.2            | 1023.4               | 20.31                |
> | Diff **+Lighter Wave**       | **2644.2**        | **617.6**           | **9.14**             |
>
> **Notes**
>
> - **Full Wave (original proposal).**
>   Implements a discrete **Hamiltonian** form of the wave equation by introducing an explicit _velocity_ state plus Velocity-FFN and Velocity-LN. While it raises accuracy on some tasks, the extra state and auxiliary blocks slow both inference and training, as shown above.
>
> - **Lighter Wave (second-order, momentum-only).**
>   Responding to the reviewer’s cost concern, we removed the auxiliary Velocity-FFN/LN and kept only the momentum term
>   $$
>   x_{l+1}=x_l+\operatorname{attn}(x_l)+\beta\bigl(x_l-x_{l-1}\bigr),
>   $$
>   where $\beta$ is the mixing parameter. This preserves the physical insight, yet matches baseline throughput (NLP -0.30 %, CV +0.5 %) and memory (NLP +2.1 %, CV +10.7 %, still < 1 GB).
>
> - **Effect on deep graphs.**
>   Applying **Lighter Wave** to 20-layer citation-graph models still _vastly_ outperforms diffusion (+45 pt on Cora, +20 pt on PubMed; see Table 2).
>
> > Guided by the reviewer’s feedback, we now present **Lighter Wave** as the _practical option_: it preserves our physical motivation while keeping compute overhead within a few percent. We thank the reviewer for inspiring this simplification.
>
> ---
>
> # 4. Responses to the reviewer’s questions
>
> |#| Question (paraphrased)                                                | Answer                                                                                                                                                                                                                                                                                                               |
> |--| ----------------------------------------------- | ---------------------------------------------------------------------------------------------------------------------------------------------------------------------------------------------------------------------------------------------------------------------------------------------------------------------------- |
> |**Q1**| How does the method scale to larger models? | On a 24-layer CaIT-XXS and citation graphs, wave update boosts accuracy by **+1 pt** on ImageNet and up to **+45 pt** on graphs respectively without tuning. Because the update is local to each block, scaling hidden size or depth follows the same complexity curve as a standard Transformer. |
> |**Q2**| Runtime/memory overhead?                    | See Tables 4 and 5.                                                                                                                                                                                                                                                                                                                |
>
> ---
>
> # 5 Planned manuscript updates
>
> - **Experiments** – add comparison with CaIT-XXS and the results of graph benchmarks including the trends of cosine similarity for each case of Cora/Citeseer/Pubmed.
> - **Cost analysis** – insert full table for computational cost with the lighter wave variant.
> - **Method** - add formalization of the lighter wave update.
>
> * All numerical details and training scripts will be released upon acceptance.
>
> ---
>
> ## Remark — Ongoing fine-tuning experiments (GPT-2 / GPT-2-Large)
>
> We are currently fine-tuning **GPT-2 (124 M)** and may also fine-tune **GPT-2-Large (774 M)** on the OpenWebText corpus, comparing the diffusive baseline with our wave update. Experiments are ongoing; full results will be reported in the camera-ready.
>
> ---
>
> We hope these additions fully address your questions and demonstrate that the proposed wave dynamics are both scalable and efficient.
>
> _Thank you again for your encouraging review._

---

### Note · Authors · 2025-08-12

Thank you to the AC and reviewers for the constructive discussions.

**Core contribution.** We reinterpret stacked attention as diffusion on a complete graph, explaining over-smoothing as a dissipative effect. We replace it with a second-order wave residual—**Wavy Transformer**—and align FFN/LN with the state–velocity relation. Across NLP and vision, we observe consistent gains with minimal extra parameters and no extra tuning.

**Key results added during review**
- **Wave-residual gains**
  - ImageNet-1K (CaIT-XXS-24): **+1.0 Top-1** with **−0.9M params**.
  - Deep citation graphs: **Cora (20 layers) +45.26**, **PubMed (20 layers) +20.54**.
  - OGB: **OGBN-Arxiv (7 layers) +42.29**, **OGBN-Proteins (5 layers) +10.72 AUC**.
- **Computational cost**
  - Full Wave increases cost; **Light Wave** matches baseline throughput/memory: **BERT 101.6→101.3 k tokens/s**, **DeiT 2631→2644 img/s**.
  - Accuracy preserved or improved: **GLUE 65.43→66.27**, **DeiT-Tiny 72.17→73.09**.
- **Over-smoothing**
  - Graphs: diffusion drives cosine similarity to **≈1.00**; Wavy stays **≤0.38** across depth.
  - Vision: spectral gap **0.836→0.629**; node-feature variance increases; **Light Wave** shows the largest inter-class variance.

**Positioning.** We added the missing citations and clarified scope. Prior work either adds wave dynamics to sparse-graph GNNs or enforces Hamiltonian constraints. In contrast, we replace the internal dynamics of complete-graph attention in the Transformer; to our knowledge, this is the first wave update inside a Transformer with complete-graph attention.

**Reinforcement, not revision.** The core method and claims are unchanged; the new experiments strengthen and clarify them. **Light Wave** follows the same derivation and serves as a practical variant. Our conclusions remain the same.

**Summary.** We believe all concerns are addressed. The method is unchanged. The new results and clarifications reinforce the original contribution. We hope this concise summary helps the AC in the final decision.

**Quick verification.**

*Please consider whether these updates address the concerns.*
- **YBPV:** Concerned that revisions require a full reread; the method is unchanged and based on a consistent physical basis; additions are supporting evidence only.
- **pC7f:** Concerned about missing related work and small/dated graph benchmarks; the missing related work is cited and OGB results were added.

---

### Decision · Program_Chairs · 2025-09-17

**Decision:**

Accept (poster)

**Comment:**

It has been known that GNNs implement a form of dissipative (heat) diffusion dynamics that can result in spatial over-smoothing.  The use of (oscillatory) wave equations has also been explored to mitigate over-smoothing, but the authors extend the paradigm to all-to-all attention based transformer networks.  The modification results in better performance in NLP and vision tasks.

The paper is motivated and written with excellent clarity.
The proposed method is likely not the last word on reducing information loss in graph neighborhood compression, but may be a good tool in the graph representation toolbox.
Reviewers YBPV and pC7f have helped ground the work better wrt prior work.
All reviewers helped motivate the authors to put in a terrific amount of effort during rebuttal, leading to more experiments and a much better understanding of the results.

Even after allowing for the discount applied by some reviewers to the originality and impact, the paper would srike up interesting conversation if presented as a poster.